

# Airborne observation with a low-cost hyperspectral instrument: Retrieval of NO₂ VCD and the satellite sub-grid variability over industrial point sources

Jong-Uk Park[1], Hyun-Jae Kim[2], Jin-Soo Park[2], Jinsoo Choi[2], Sang Seo Park[3], Kangho Bae[3], Jong-Jae
Lee[3], Chang-Keun Song[3], Soojin Park[1], Kyuseok Shim[1], Yeonsoo Cho[1], and Sang-Woo Kim[1]

[1] School of Earth and Environmental Sciences, Seoul National University, Seoul, Korea
[2] Climate and Air Quality Research Department, National Institute of Environmental Research, Incheon, Korea
[3] Department of Urban and Environmental Engineering, Ulsan National Institute of Science and Technology, Ulsan, Korea

*Correspondence to*: Sang-Woo Kim (sangwookim@snu.ac.kr)

**Abstract.** The high spatial resolution NO₂ vertical column densities (VCDs) were measured from the airborne observations using the low-cost Hyperspectral Imaging Sensor (HIS) at three industrial areas (i.e., Chungnam, Jecheon, and Pohang) in Korea, where point sources (i.e., power plant, petrochemical complex, steel yard, and cement Kiln) with significant NO₂ emissions are located. An innovative and versatile approach for NO₂ VCD retrieval, hereafter referred to as the Modified Wavelength Pair (MWP) method, was developed to overcome the excessively variable radiometric and spectral characteristics of the HIS attributed to the absence of temperature control during the flight. The newly developed MWP method was designed to be insensitive to broadband spectral features, including the spectral dependency of surface and aerosol reflectivity, and can be applied to observations with relatively low spectral resolutions. Moreover, the MWP method can be implemented without requiring precise radiometric calibration of the instrument (i.e., HIS) by utilizing clean pixel data for non-uniformity corrections and is also less sensitive to the optical properties of the instrument and offers computational cost competitiveness. In the experimental flights using the HIS, NO₂ plumes emitted from steel yards were particularly conspicuous among the various NO₂ point sources, with peak NO₂ VCD of 2.0 DU at Chungnam and 1.8 DU at Pohang. The typical NO₂ VCD uncertainties ranged between 0.025–0.075 DU over the land surface and 0.10–0.15 DU over the ocean surface, and the discrepancy can be attributable to the lower signal-to-noise ratio over the ocean and higher sensitivity of the MWP method to surface reflectance uncertainties under low-albedo conditions. The NO₂ VCDs retrieved from the HIS with the MWP method showed a good correlation with the collocated TROPOMI data (R=0.73, mean absolute error=0.106 DU). However, the temporal disparities between the HIS frames and the TROPOMI overpass, as well as the different observation geometries under complex vertical wind fields, limited the correlation. The comparison of TROPOMI and HIS NO₂ VCD further demonstrated that the satellite sub-grid variability could be intensified near the point sources, with more than a threefold increase in HIS NO₂ VCD variability (e. g., difference between 25th and 75th quantiles) over the TROPOMI footprints with NO₂ VCD values exceeding 0.8 DU compared to footprints with NO₂ VCD values below 0.6 DU.



## 1 Introduction

Reactive nitrogen oxides (NOx), which is a sum of nitrogen dioxide ($NO_2$) and nitrogen monoxide (NO), are one of the most
important trace gases in a polluted atmosphere. NOx has been widely known to have direct health effects (Jo et al., 2021;
Kampa and Castanas, 2008; Song et al., 2023) and takes an essential role in secondary ozone ($O_3$) and aerosol formation
(Chan et al., 2010; Seinfeld and Pandis, 2006; Sillman, 1999), both of which are also the major atmospheric pollutant. NOx
can be emitted from natural sources such as wildfires or lightning (Jaffe and Wigder, 2012; Stark et al., 1996), whereas
anthropogenic emissions, mainly from the combustion of fossil fuels, are the main cause of the air quality deterioration in
populated urban and industrial regions (Kim et al., 2013; Lamsal et al., 2013). NOx is highly reactive in the polluted
atmosphere and exhibits large spatial inhomogeneity and short atmospheric lifetime (Beirle et al., 2011; Liu et al., 2016;
Valin et al., 2013). Therefore, surface in-situ measurements of speciated NOx must be complemented by remote sensing
observations with wider spatial coverage. $NO_2$ has a strong and highly structured absorption cross-section at the visible
(VIS) band, and the column densities can be retrieved from passive hyperspectral observations and with various spectral
analysis techniques such as Differential Optical Absorption Spectroscopy (DOAS; Platt and Stutz, 2008). Considering the
balanced state of NO and $NO_2$ from NOx titration and photolysis during the daytime (Sillman, 1999), observations of $NO_2$
concentrations can yield the general concentration of NOx in the atmosphere.

Since the first global $NO_2$ vertical column density (VCD) observations from push-broom type satellite hyperspectral imaging
sensor, the Global Ozone Monitoring Experiment (GOME; Burrows et al., 1997), numerous hyperspectral imagers such as
Ozone Monitoring Instrument (OMI; Levelt et al., 2006), GOME-2 (Munro et al., 2016), and Tropospheric Monitoring
Instrument (TROPOMI; Veefkind et al., 2012) were sequentially launched with gradual enhancement in its spatial
resolution. For instance, TROPOMI, the latest sensor on the low-Earth orbit, has a footprint of $3.5 \times 5.5$ km$^2$ at the nadir,
while the GOME had a footprint of $40 \times 320$ km$^2$. However, even up-to-date environmental satellite sensor such as
TROPOMI shows inherent limitations in spatial resolution to fully capture the $NO_2$ spatial inhomogeneities in urban and
industrial areas (Judd et al., 2019; Park et al., 2022; Tang et al., 2021; Verhoelst et al., 2021).

The airborne observation of $NO_2$ VCDs from hyperspectral imagers has demonstrated its effectiveness in elucidating the
spatial variability of $NO_2$ near emission hot spots (Judd et al., 2019; Tack et al., 2021). However, the highly resolved
airborne observations of $NO_2$ VCDs were sparse and limited to intensive field campaigns. State-of-the-art hyperspectral
imagers dedicated to observing column densities of trace gases such as the Geostationary Trace gas and Aerosol Sensor
Optimization (GeoTASO; Leitch et al., 2014), GEO-CAPE Airborne Simulator (GCAS; Kowalewski and Janz, 2014),
Airborne imaging DOAS instrument for Measurements of Atmospheric Pollution (AirMAP; Schönhardt et al. 2015), and
Airborne Compact Atmospheric Mapper (ACAM; Kowalewski and Janz, 2009; Lamsal et al., 2017), have been widely
deployed in intensive field campaigns. Although these hyperspectral radiometers provide highly accurate measurements
owing to their stable spectral-radiometric characteristics and precise spectral resolution, these instruments are expensive to
develop and require sophisticated and meticulous calibrations and maintenance to retain their performance.



Meanwhile, efforts have been made to retrieve NO$_2$ VCDs from airborne observations using hyperspectral imagers that were not originally designed for trace gas observations. Popp et al. (2012) and Tack et al. (2017; 2021) used Airborne Prism Experiment (APEX; Itten et al., 2008), which is a hyperspectral imager initially developed to observe land use-land cover (LULC; Tack et al., 2019), to retrieve NO$_2$ VCDs in various airborne experiments. The APEX instrument has a variable and

coarse spectral resolution in the VIS band with full width at half maximum (FWHM) ranging from 1.5 to 3.0 nm. Nevertheless, the NO$_2$ VCDs were successfully retrieved by applying conventional DOAS fitting and air mass factor (AMF) calculations, with a good agreement to collocated ground-based DOAS instruments (R=0.84) and TROPOMI (R=0.94; APEX data upscaled to the TROPOMI footprints).

To enhance our understanding and obtain explicit depictions of atmospheric pollutants, improving the feasibility of airborne

observations is important. The value of airborne NO$_2$ VCD observations can be further highlighted in East Asia, where the NOx emissions are still substantial, and tropospheric O$_3$ levels are increasing (Colombi et al., 2023; Li et al., 2020). However, there are challenges associated with airborne hyperspectral observations, including the development of sophisticated instruments and the need for precise calibrations and maintenance. Therefore, in this study, we developed a versatile NO$_2$ VCD retrieval algorithm suitable for a broader range of hyperspectral instruments. The newly developed

algorithm has been implemented to spectra obtained from experimental flights with a low-cost hyperspectral imaging sensor, and the associated retrieval uncertainties were estimated. Furthermore, retrieved NO$_2$ VCDs were compared with the spatially collocated TROPOMI data, focusing on the satellite sub-grid variability observed over industrial point sources in South Korea.

## 2 The low-cost Hyperspectral Imaging Sensor (HIS)

The compact, low-cost hyperspectral imaging sensor (hereafter referred as HIS) designed to observe from ultraviolet (UV) to VIS bands (250 – 500 nm) in hyperspectral resolution was used in this study for airborne observations. Table 1 shows the detailed specifications of the HIS, manufactured by Headwall Photonics, Inc. The HIS has a 2-D 16-bit Charge-Coupled Device (CCD), which supports relatively large dynamic range of digital photon counts from 0 to $2^{16}$, with 1,392 (spatial columns) $\times$ 1,040 (spectral rows) arrays as a detector. The CCD conversion rate to digital count as a response to incident

photons is known to be non-linear when the CCD counts exceed approximately 80 % of the saturation level, and the CCD counts should remain below 50,000 counts for practical use. Furthermore, the latest (spectral) rows of the CCD are designed not to be illuminated, yielding 945 effective spectral rows. A diffraction grating is used with a concave mirror to disperse light as a spectrum, and a 25 μm-width slit on the light entrance is designed to yield a slit function with 1.4 nm FWHM. The HIS has a 0.265 nm spectral binning interval on average and a moderate level of oversampling (approximately 5 times)

accordingly.



**Table 1. Specifications of the HIS used in this study.**

| Specifications | Values | Remarks |
|---|---|---|
| Wavelength range | 250 – 500 nm | 945 effective spectral pixels |
| Spectral binning interval | 0.265 nm | - |
| Spectral resolution | 1.4 nm | FWHM |
| Spatial pixels | 1,392 | - |
| FOV | 13 ° | Varying with the focal length of the lens |
| Detector | Interline CCD | 16-bit |
| Slit width | 25 $\mu m$ | - |
| Lens EFL | 28.3 mm | The same objective lens was used throughout the study |

Unlike the instrument specifications, the HIS shows almost no sensitivity beyond the UV-A region ($\lambda < 320$ nm). On the other hand, the HIS exhibits sufficient sensitivity at the VIS range (350–500 nm), where the spectral absorption characteristics of $NO_2$ are well distinguished, making it suitable for $NO_2$ VCD retrievals. For instance, Park et al. (2019) retrieved $NO_2$ VCD from ground-based observations of the diffuse radiances using the identical instrument (i.e., HIS) and presented its compatibility for $NO_2$ VCD retrievals by demonstrating a high correlation (R=0.84) with collocated Pandora

observations. However, there are several vulnerabilities on the HIS that hinder its utilization for airborne observations, primarily due to low-quality optical and radiometric characteristics resulting from its compact configuration and lack of regular or meticulous maintenance.

First and foremost, the HIS is not equipped with a temperature control unit, which is crucial to ensure ground calibration to preserve its validity and to keep optical and radiometric properties stable. Exposure to fluctuating ambient temperatures

during the flight can cause thermal shifts in the instrument's optics, resulting in spectral shifts or changes in the slit function. Furthermore, the sensitivity of the CCD pixels can vary, even from pixel to pixel, with temperature changes, leading to additional pixel non-uniformity. The HIS inherently exhibits relatively unstable spectral and radiometric characteristics compared to the sophisticated state-of-the-art hyperspectral imagers like GeoTASO or GCAS, which is another drawback. Additionally, the narrow field of view (FOV) of the HIS can be a critical defect when used for airborne observations,

resulting in a narrow swath. Theoretically, the FOV can be enlarged by using an objective lens with a shorter focal length. However, the objective lens with an effective focal length (EFL) of 28.3 mm used in this study is already compact with an excessively short focal length (i.e., low f-number), and is not feasible to enlarge the field of view under current instrument configuration.



## 3 NO₂ VCD retrieval algorithm

Most modern passive remote sensing techniques aimed to observe atmospheric trace gases (e.g., NO₂, O₃, formaldehyde) in UV–VIS bands via hyperspectral sensors adopt the DOAS technique, which necessarily requires sufficiently high spectral resolution and well-defined, stable optical characteristics (e.g., slit function). More specifically, DOAS fitting technique is applied to the observed spectra from airborne or space-borne hyperspectral images to retrieve slant column densities (SCDs), and by dividing with the AMF calculated from the forward radiative transfer model (RTM) and the chemical transfer model

(CTM) yields VCDs. To follow the convention and retrieve VCDs accurately, precise spectral and radiometric characterization before and after each research flight, as well as stabilization in terms of temperature and vibration during the flight, are necessary. However, the demand for precise calibration and the need for ancillary compartments, such as temperature control units, pose obstacles that impede the feasibility of airborne hyperspectral observation. Therefore, we have developed an innovative and versatile approach for NO₂ VCD retrieval, hereafter referred to as the Modified

Wavelength Pair (MWP) method, which can be applied to spectra obtained from various hyperspectral instruments. Figure 1 illustrates the overall flowchart outlining the NO₂ VCD retrieval from the airborne HIS spectra, including the implementation of the newly developed MWP method. Each step of the flowchart is elaborated in the following sections.

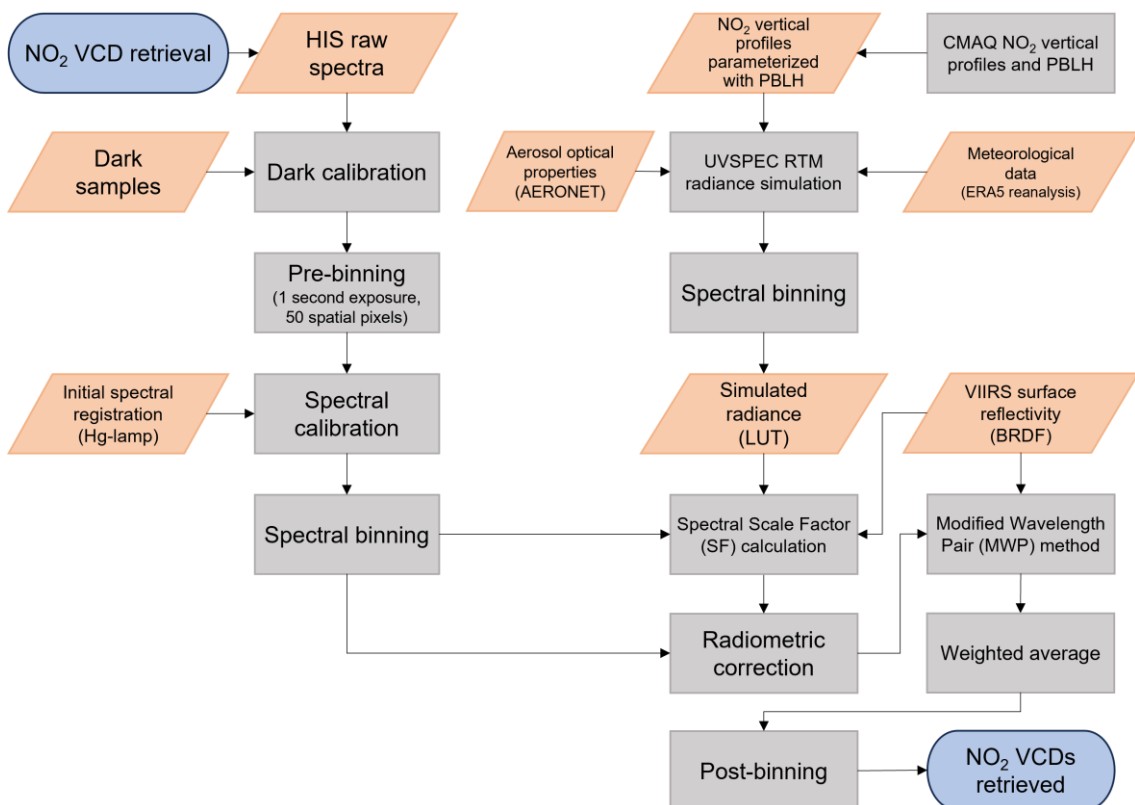

**Figure 1. Flowchart of the NO₂ VCD retrieval from the airborne HIS spectra using the MWP method.**



## 3.1 Theoretical basis of the Modified Wavelength Pair (MWP) method

A total of six wavelength pairs have been selected within the fitting window ranging from 410–450 nm, where the $NO_2$ absorption exhibits significant high-frequency spectral variability within the VIS band. These pairs are composed of wavelengths that exhibit weak and strong $NO_2$ absorption and are classified into two sub-types: Type_A and Type_B, with

three wavelength pairs allocated to each type (Table 2). In Type_A pairs, the shorter wavelength $\lambda_1$ shows stronger sensitivity to $NO_2$ concentration, while the longer wavelength $\lambda_2$ exhibits relatively weaker $NO_2$ sensitivity (Fig. 2). In contrast, Type_B pairs show the opposite pattern, with weaker $NO_2$ sensitivity at the shorter wavelength ($\lambda_1$) and stronger sensitivity at the longer wavelength ($\lambda_2$). The basic concept of using the wavelength pair is that the radiance ratio ($R = I_{\lambda_1}/I_{\lambda_2}$) calculated from the spectral radiance at two different wavelengths of a wavelength pair, $\lambda_1$ and $\lambda_2$ (where $\lambda_1 < \lambda_2$),

is expected to be pseudo-linearly dependent on the $NO_2$ VCD. Therefore, $NO_2$ VCD can be retrieved by comparing the observed $R$ value with the $R$ values simulated from the RTM for a range of possible $NO_2$ VCDs (and controlling other possible artifacts).

**Table 2. Wavelength-pairs used in NO₂ VCD retrieval algorithm (MWP method).**

| MWP Type | Wavelength pair index | Wavelength 1 ($\lambda_1$) | Wavelength 2 ($\lambda_2$) |
|:---:|:---:|:---:|:---:|
|   | 1 | 414.209 nm | 415.535 nm |
| **A** | 2 | 435.689 nm | 437.015 nm |
|   | 3 | 439.932 nm | 441.258 nm |
|   | 1 | 417.126 nm | 418.452 nm |
| **B** | 2 | 433.037 nm | 434.363 nm |
|   | 3 | 442.849 nm | 444.175 nm |


Multiple wavelength pairs are selected within the fitting window to reduce the uncertainty. However, when wavelength pairs following the convention of only Type_A or Type_B are selected, systematic biases in the retrieved $NO_2$ VCDs can occur if the assumed surface and aerosol reflectance spectral dependency in the RTM simulation differs from the real world. For example, when the wavelength pairs are only comprised of Type_A, the $R$ value for the wavelength pair is expected to be

negatively correlated with the $NO_2$ VCD. In conditions where the reflectance is higher at the longer wavelength, which is common for land or vegetated surface, the $R$ value is likely to be underestimated, resulting in a positively biased $NO_2$ VCD. On the other hand, conditions with lower reflectance at a longer wavelength, which is a general feature over water, lead to an overestimation of the $R$ value and an underestimation of the $NO_2$ VCD. To mitigate these biases, it is necessary to consider accurate and highly resolved spectral surface reflectance and the aerosol optical property data, temporally and spatially

collocated to the research flights. However, acquiring such data is often challenging, and even with massive computational consumption (i.e., running high-resolution CTM), it is likely to be uncertain.





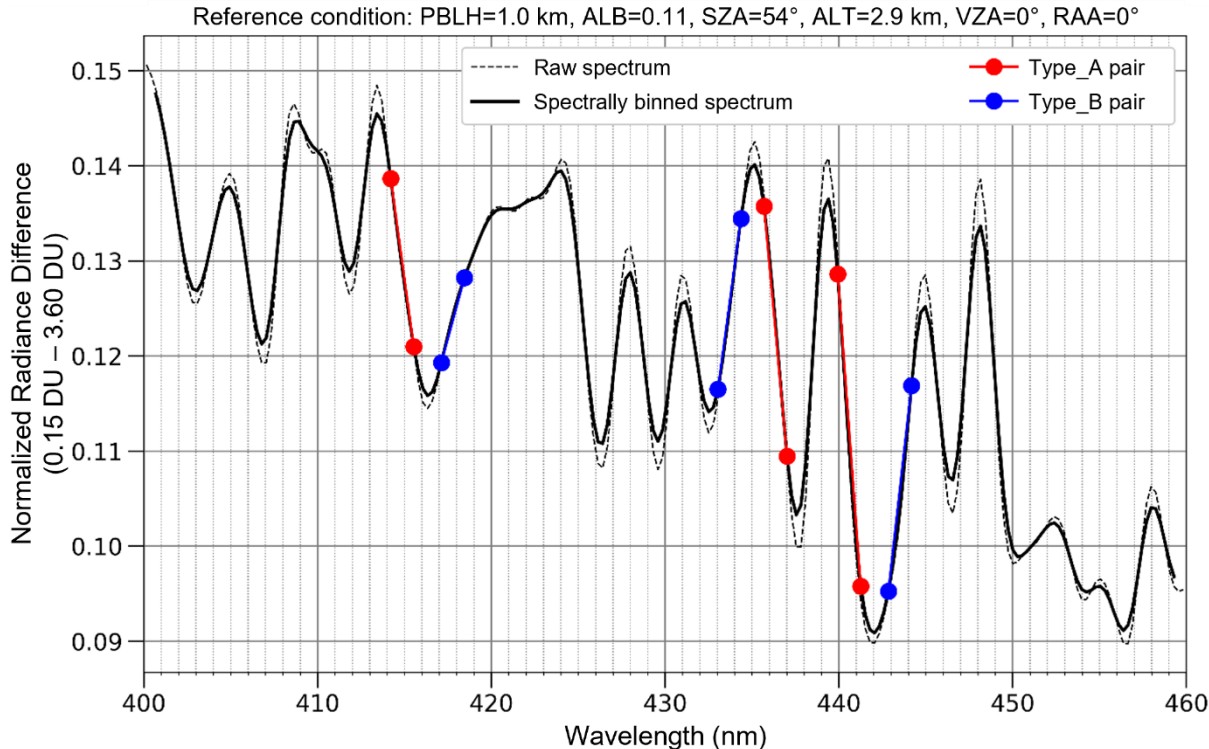

**Figure 2. Example of spectral radiance sensitivity to NO₂ VCD under reference HIS observation condition at Pohang area shown with three different wavelength-pairs per each wavelength pair type (Type_A in red and Type_B in blue) used in the MWP method.**

Given the limitations in obtaining suitable surface and aerosol data, an analytical solution has been derived aiming to minimize the influence of the spectral dependency of surface reflectivity and aerosol optical properties. For a single analytical solution (i.e., NO₂ VCD), a set of wavelength pairs comprised of one Type_A and one Type_B wavelength pair was used. Therefore, three independent analytic solutions from three sets of wavelength pairs, yield three NO₂ VCD values from a single observed spectrum. Specifically, a set of wavelength pairs were formed by grouping the neighboring wavelength pairs with the same index (shown in Table 2).

$$
\begin{cases}
VCD_A = R_{rtm,A} \cdot a_A + b_A, & a_A < 0 \ \ (\because VCD\uparrow \ \rightarrow R\downarrow) \\
VCD_B = R_{rtm,B} \cdot a_B + b_B, & a_B > 0 \ \ (\because VCD\uparrow \ \rightarrow R\uparrow)
\end{cases}
\tag{1}
$$

Equation (1) shows the relationship between NO₂ VCD (abbreviated as "VCD" in the equations for simplicity) and the *R* values of the Type_A and Type_B wavelength pairs, respectively, assuming the first-order linear relationship between the two. This assumption was confirmed from the forward RTM simulations (see details in Sect. 3.2), by the squared correlation



coefficient ($R^2$) of the regressed line (i.e., Eq. 1) exceeding 0.99 (shown in supplementary Fig. S1) for all six wavelength

pairs.

As we assume the ratio between real-world reflectivity (i.e., from the surface and aerosols) and the assumed reflectivity for

the RTM at $\lambda_2$ is a k-fold of the value at $\lambda_1$, the relation between the $R$ value of the wavelength pair simulated from the RTM

and the expected $R$ value calculated from real-world HIS observation can be expressed as Eq. (2). The same $k$ value can be

assumed for the wavelength pairs grouped as the same wavelength pair set, since the difference between the wavelengths of

the wavelength pair ($\Delta\lambda = \lambda_2 - \lambda_1$) is small (< 1.5 nm) and almost consistent at all six wavelength pairs. Moreover, the

wavelengths constituting a single wavelength pair set ranges within 5 nm bandwidth, so we can assume the spectral

dependency to be invariant.

$$\begin{cases} R_{obs,A} = \frac{1}{k} \cdot R_{rtm,A} \\ R_{obs,B} = \frac{1}{k} \cdot R_{rtm,B} \end{cases} \tag{2}$$

Here, subscript "rtm" and "obs" refers to the simulated value from the RTM and the expected value from the real

hyperspectral observations, respectively. From Eqs. (1) and (2), the relation between the biased VCD estimates ($VCD_{obs}$) and

the observed radiance ratio of the wavelength pair ($R_{obs}$), can also be expressed with the simulated radiance ratio ($R_{rtm}$)

together with the spectral dependency factor $k$ (Eq. 3).

$$\begin{cases} VCD_{obs,A} = R_{rtm,A} \cdot \frac{a_A}{k} + b_A = R_{obs,A} \cdot a_A + b_A \\ VCD_{obs,B} = R_{rtm,B} \cdot \frac{a_B}{k} + b_B = R_{obs,B} \cdot a_B + b_B \end{cases} \tag{3}$$

Since the simulated NO$_2$ VCDs should be the same regardless of wavelength pair type, the "True" NO$_2$ VCD ($VCD_{True}$) can

be expressed as follows (Eq. 4).

$$VCD_{True} = VCD_{rtm,A} = VCD_{rtm,B} = R_{rtm,A} \cdot a_A + b_A = R_{rtm,B} \cdot a_B + b_B \tag{4}$$

From Eq. (4), $R_{rtm,A}$ can be shown as Eq. (5).

$$R_{rtm,A} = \frac{R_{rtm,B} \cdot a_B + b_B - b_A}{a_A} \tag{5}$$

Then the Eq. (2) can be reformulated by replacing $R_{rtm,A}$ with Eq. (5), yielding Eq. (6).




$$VCD_{obs,A} - b_A = \frac{R_{rtm,B} \cdot a_B + b_B - b_A}{k} \tag{6}$$

Equations (3) and (6) can be reorganized to Eq. (7) and then be solved to a single equation (Eq. 8).

$$\begin{cases} \frac{1}{k} = \frac{VCD_{obs,A} - b_A}{R_{rtm,B} \cdot a_B + b_B - b_A} \\ \frac{1}{k} = \frac{VCD_{obs,B} - b_B}{R_{rtm,B} \cdot a_B} \end{cases} \tag{7}$$


$$\frac{VCD_{obs,A} - b_A}{R_{rtm,B} \cdot a_B + b_B - b_A} = \frac{VCD_{obs,B} - b_B}{R_{rtm,B} \cdot a_B} \tag{8}$$

Equation (8) can be reorganized as an expression upon $R_{rtm,B}$, yielding Eq. (9).


$$R_{rtm,B} = \frac{1}{a_B} \cdot \frac{b_A - b_B}{VCD_{obs,A} - b_A} \cdot \frac{1}{\left( \frac{1}{VCD_{obs,A} - b_A} - \frac{1}{VCD_{obs,B} - b_B} \right)}$$

$$= \frac{1}{a_B} \cdot \frac{(b_A - b_B) \cdot (VCD_{obs,B} - b_B)}{VCD_{obs,B} - VCD_{obs,A} + b_A - b_B} \tag{9}$$

Then the final equation, Eq. (10), for the "True" unbiased NO$_2$ VCD is derived only as a function of the observed $R$ values of the Type_A and Type_B wavelength pairs ($R_{obs,A}$, $R_{obs,B}$) and the coefficients for the first order polynomials regressed

upon the relationship between simulated $R$ values and the NO$_2$ VCDs (i.e., $a_A, b_A, a_B,$ and $b_B$ in Eq. 1).

$$VCD_{True} = R_{rtm,B} \cdot a_B + b_B = \frac{(b_A - b_B) \cdot (VCD_{obs,B} - b_B)}{VCD_{obs,B} - VCD_{obs,A} + b_A - b_B} + b_B = \frac{(b_A - b_B) \cdot (R_{obs,B} \cdot a_B)}{R_{obs,B} \cdot a_B - R_{obs,A} \cdot a_A} + b_B \tag{10}$$

As shown in Eq. (10), in which the NO$_2$ VCD is expressed without the broadband spectral dependency term ($k$), the NO$_2$

VCDs can be retrieved with the MWP method without considering the varying spectral variability of the surface or the aerosol properties. For the MWP method application, a set of wavelength pairs, one Type_A and one Type_B, is required. Therefore, from three sets of wavelength pairs, three independent NO$_2$ VCD values can be retrieved from a single observed spectrum. It is noteworthy to mention that the spectral binning was applied to the HIS observed spectra and subsequently the RTM simulated radiance spectra before implementing the MWP method, aiming to increase the signal-to-noise ratio (SNR).

The spectral binning was constrained within a range that does not sacrifice the sensitivity of the wavelength pair method (Fig. 2) and takes into account the HIS oversampling rate, resulting in spectral binning of ± 2 original spectral bins.





## 3.2 Forward radiative transfer modeling

In order to apply MWP method to the observed HIS spectra, a Look-Up Table (LUT) of the *R* values corresponding to different NO₂ VCDs under various environments and observation conditions should be constructed from the forward RTM
simulations for each wavelength pairs. The UVSPEC, specialized in radiative transfer calculations in UV–VIS bands, within the versatile libRadtran (version 2.0.4) radiative transfer software package (Emde et al., 2016; Mayer and Kylling, 2005) was utilized for the forward radiative transfer simulations. Spectral radiances were simulated from the UVSPEC based on input meteorological parameters and the profiles of various atmospheric constituents such as trace gases and aerosols.

**Table 3. Variables and corresponding entries for the radiative transfer model simulations to construct the LUT.**

| Variables (Acronym) | Entry range (Number of entries) | Unit | Remarks |
|---|---|---|---|
| NO₂ VCD | 0.15 – 3.60 (70) | Dobson Unit (DU) | 1 DU=2.687×10¹⁶ molecules cm⁻² |
| Planetary Boundary Layer Height (PBLH) | 0.2 – 1.8 (5) | km | - |
| Surface Reflectivity (ALB) | 0.01, 0.03, 0.05, 0.08, 0.11, 0.15, 0.20, 0.25, 0.30, 0.35 (10) | - | Assumed as Lambertian surface without spectral dependency |
| Solar Zenith Angle (SZA) | Variable per flight | Degree (°) | < 70° |
| Flight Altitude (ALT) | Variable per flight | m | Above the mean sea level |
| Viewing Zenith Angle (VZA) | 0 – 24 (9) | Degree (°) | - |
| Relative Azimuth Angle (RAA) | 0 – 180 (7) | Degree (°) | - |
| Wavelength | 400.15 – 459.82 (226) | nm | encompassing the fitting window |

Table 3 shows the input conditions to the UVSPEC for the LUT construction. Some of the input entries were adjusted to each research flight to account for possible conditions that vary in location and time. The ECMWF Reanalysis v5 (ERA5)
hourly reanalysis data (Hersbach et al., 2020; 2023a) over the corresponding flight area and time of each research flight was used as an input atmospheric condition (i.e., vertical profiles of pressure, temperature, and mixing ratio of major atmospheric gaseous constituents except for NO₂) for the RTM simulations. The vertical profile of NO₂ can significantly alter the AMF, or the effective absorption of NO₂ per unit VCD, so it is necessary to use the legitimate profile as possible. The most intuitive way is to use vertical profiles from highly resolved CTM data. However, unlike conventional AMF calculation
using LUT of scattering weights and taking the inner product with CTM simulated vertical profiles, it is not feasible to take all the vertical profiles within the target domain into account and construct the LUT of the simulated radiances, of which the MWP method requires.



As an alternative, $NO_2$ vertical profiles from the high-resolution (3 km × 3 km horizontal resolution; 23 vertical grid) Community Multiscale Air Quality (CMAQ) v5.2 model (Appel et al., 2017; US EPA, 2017) simulations at each target

domain were collected to calculate representative $NO_2$ vertical profile under certain Planetary Boundary Layer Height (PBLH) and $NO_2$ VCD conditions. The GMAP/SIJAQ v2.0 emission inventory based on the Clean Air Policy Support System (CAPSS) 2018 and Comprehensive Regional Emissions inventory for Atmospheric Transport Experiment (CREATE) v5.0 (Woo et al., 2020) under the framework of SMOKE-Asia (Woo et al., 2012) emission model was used as a high-resolution (3 km × 3 km) anthropogenic emissions for the CMAQ, while the Model of Emissions of Gases and Aerosols

from Nature (MEGAN) v2.10 was used as a natural emissions. Meteorological parameters were simulated from high-resolution (3 km × 3 km horizontal resolution; 58 vertical grid) Weather Research and Forecasting (WRF) v3.9.1 model (Skamarock et al., 2008), with the initial conditions from the National Centers for Environmental Prediction/Final Analysis (NCEP/FNL) reanalysis data. More specifically, $NO_2$ vertical profiles within each target domain were grouped by the CMAQ-driven PBLH with ± 0.2 km range from the PBLH entries shown in Table 3. The variability (i.e., quantiles) of $NO_2$

mixing ratio at each altitude level was calculated from vertical profiles grouped under the same PBLH conditions, and the $NO_2$ vertical profile corresponding to certain $NO_2$ VCD was deduced from Levenberg-Marquardt fitting upon quantile value that best fits the targeted column-integrated $NO_2$ VCD (supplementary Fig. S2). It should be noted that the stratospheric $NO_2$ VCD (i.e., $NO_2$ column densities above 15 km in altitude) was assumed constant (0.185 DU) based on the US standard atmosphere (1976), since the CMAQ only spans altitudes up to approximately 70 hPa.

Appropriate consideration of the real-world aerosol vertical profile is as important as accurately depicting the $NO_2$ vertical profile because the aerosol distribution compels the effective light path (or the scattering weight). However, due to the lack of a vertically-resolved observation dataset, column-integrated aerosol optical properties such as Aerosol Optical Depth (AOD) and Single Scattering Albedo (SSA) from the nearest AERONET (Holben et al., 1998; Kim et al., 2007; supplementary Table S1) sites were used for the radiative transfer calculation.

For the forward radiance simulation with the UVSPEC, the TSIS-1 hybrid solar reference spectrum (Bak et al., 2021; Coddington et al., 2021) was used for the solar irradiance spectrum, while the surface was assumed to exhibit Lambertian reflectance without spectral dependency. Input spectra such as absorption cross-sections of trace gases and the solar irradiance spectrum were convolved by using the Gaussian slit function with FWHM of 1.4 nm to ensure conformity with the expected observed spectra from the HIS.

**3.3 Spectral and radiometric calibrations**

Accurate spectral and radiometric characterization of an instrument is essential to assure consistency and minimize the uncertainty of the retrieved $NO_2$ VCDs. Therefore, the spectral and radiometric properties of the HIS have been assessed, as well as post-calibrations upon the obtained spectra from the airborne HIS observations. First, the SNR of the HIS was examined based on dark samples (i.e., HIS observed spectra without light exposure) collected with diverse exposure times.

Dark signals (CCD signals from the dark samples) are composed of dark current – thermally induced false signal, and the





offset value – an arbitrary value added by the manufacturer to avoid unintended positive bias from instrument noise. Dark signals had almost no dependency on the exposure time, inferring minimal dark currents of the HIS, and the dark offset ranged from 380 to 410 counts by spatial column to column. However, the noise level of the pixels of interest (spectral rows within the fitting window used in the MWP method) was substantial, necessitating additional binning to increase the SNR by

controlling the noise. Since the optimal spectral binning has already been determined and addressed in the MWP algorithm (discussed in Sect. 3.1), 50 consecutive CCD spatial columns were binned to a single co-added column, starting from the centerline column, yielding 27 co-added spatial columns and discarding 21 raw lateral columns on each side. The noise level estimated after the spatial binning was approximately 2–2.5 counts under an integration time of 1 second (supplementary Fig. S3), which would result in an SNR ranging in the scale of a few thousand (depending on the signal intensity) during the

airborne observations. Spatial binning was implemented to the HIS-observed spectra and has been taken into account for all the subsequent calibrations below.

The spectral calibrations of the instrument and the observed spectra from the HIS include (1) Slit function characterization, (2) initial spectral registration for spatially binned CCD pixels, and (3) the retrieval of wavelength shifts from the initial spectral registration for every observed frame and spatial columns. The slit function characterization and initial spectral

registration were carried out by illuminating the HIS with a reference Hg-lamp, which emits pseudo-monochromatic emission lines from the low-pressure Hg gas in the UV–VIS range. The shape of the slit function was assumed to be a simple Gaussian distribution, and the slit function FWHM was retrieved for each Hg-lamp emission line within the HIS spectral range. The slit function FWHM generally conformed to the instrument specifications (Table 1) but varied within the range of 1.1–1.4 nm. The variability of the slit function FWHM primarily arises from the deviation of the actual slit function from the

Gaussian distribution due to CCD non-uniformity and the instability in the optics (i.e., slit, grating, etc.) caused by thermal variation and vibrations. The initial spectral registration of the spatially binned CCD pixels was also accomplished by first finding the spectral row numbers corresponding to the centerline of the illuminated CCD pixels for each Hg-lamp emission line with known wavelengths. Then, the wavelengths for the rest of the spectral rows were registered by interpolating the spectral rows with known wavelengths.

The initial pre-flight wavelength registration should be calibrated since the HIS is exposed to varying temperatures during the flight, which can result in a spectral shift from the initial spectral registration. The average spectral shift within the fitting window (410–460 nm) of the MWP method was calculated for every HIS-observed frame and spatial columns by fitting the spatially binned HIS spectra with the convolved solar irradiance spectrum using Levenberg-Marquardt non-linear fitting. Polynomials for scaling and offset terms, together with spectral shift and the FWHM value of the slit function, were

considered for the fitting. The detailed descriptions of spectral calibration, including slit function characterization, initial spectral registration, and spectral shift calculation, can be found in previous papers (Kang et al., 2020; Liu et al., 2015; Nowlan et al., 2016).

In order to convert raw CCD counts into a radiance, it is necessary to obtain precise pixel-by-pixel sensitivity from meticulous facilities equipped with well-calibrated reference lamp and integrating sphere. However, the intension of using





the low-cost HIS is to make the calibration and maintenance of the instrument as simple as possible for feasible observations, which contradicts with equipping with such costly facilities. Moreover, the radiometric sensitivity of the CCD alters by time and with temperature change, and even the most accurate absolute radiometric calibration does not guarantee the integrity of the HIS observed radiance during the flight. Therefore, an alternative approach based on HIS spectra obtained from the clean pixel was used to minimize the effect of non-uniformity in CCD sensitivities.

The basic concept of clean pixel calibration is determining the actual in-flight pixel-by-pixel CCD sensitivities by comparing the HIS spectra (i.e., raw digital counts) at the clean pixel with the simulated radiance from the RTM. The spectral scale factor (SF), a spectral conversion factor corresponding to the CCD sensitivity, is calculated for every pre-binned spatial column by dividing the RTM-simulated radiance with the observed HIS raw counts. It is important to accurately account for the parameters affecting the effective light absorption of $NO_2$, such as surface reflectivity and $NO_2$ VCD, in the RTM

simulation over the clean pixel. Therefore, clean pixels were selected as a place with easily identifiable homogeneous surface conditions (i.e., ocean) and with a background level of $NO_2$ near the surface. It is also important to select a clean pixel collected after a sufficient time for instrument stabilization at the observation altitude to avoid sensitivity change due to temperature change.

Applying spectral SF as an alternative radiometric calibration can result in a substantial error in the absolute magnitude of

converted radiances. However, the MWP method uses the relative radiance ratio of the wavelength pair, and the pixel-to-pixel non-uniformity is the only primary concern, not the absolute magnitude of the conversion factor. Moreover, by applying SF calculated as a ratio between simulated and observed spectra, the discrepancies between the RTM simulation settings and the real-world, including the uncertainties in instrument characterization, can be corrected empirically. To take this advantage, it is necessary to retrieve spectral SF after accurate spectral calibration and to use respective SF for every

research flight, different flying altitude, and exposure time. Meanwhile, on the other side of the virtue of applying the spectral SF from a clean pixel, it has one critical drawback in estimating the $NO_2$ VCD value corresponding to the clean pixel. Whenever a certain $NO_2$ VCD value is determined for a clean pixel, it directly affects all the other $NO_2$ VCDs retrieved from the HIS using the spectral SF obtained with a particular clean pixel. As a result, a radiometric calibration using spectral SF yields relative $NO_2$ VCD, which may be biased proportionally to the bias of $NO_2$ VCD estimates at the

clean pixel. Nevertheless, the application of spectral SF offers more significant merits than downsides, especially for calibrating low-cost hyperspectral sensor with unstable optical-radiometric characteristics that are operated without sophisticated laboratory calibrations or maintenance, since it not only calibrates pixel non-uniformity but also addresses other uncertainties arising from the inaccurate depiction of instrumental characteristics during the flight.

### 3.4 Error estimation

Uncertainties of the $NO_2$ VCDs retrieved from the MWP method can be estimated from the errors of input parameters and their sensitivity to the resulting $NO_2$ VCD. One of the advantages of the MWP method is that the solution ($NO_2$ VCD) is



retrieved analytically, making the uncertainty estimation feasible by adopting Gaussian error propagation. The error estimation of the $NO_2$ VCDs starts from the final equation (Eq. 10) of the MWP method discussed in Sect. 3.1.

The regression parameters for the wavelength pair set (i.e., $a_A, b_A, a_B$, and $b_B$ in Eq. 10) are calculated from the $NO_2$ VCD to
$R$ value relationship, whereas the $R$ values for the regression are retrieved by interpolating the LUT to the corresponding observation conditions (i.e., SZA, VZA, RAA, PBLH, etc.). Therefore, these regression parameters can be assumed as assured values for a certain observation condition. However, when the real-world observation conditions differ from the input atmospheric condition assumed for the retrieval, the $R$ values will change, posing an error on the resulting $NO_2$ VCD. Accordingly, the sensitivity of these $R$ values as a response to the uncertainties of input parameters was calculated in order to
estimate the error propagated to the final $NO_2$ VCD retrieval in the MWP method.

To calculate error propagation, $R_{obs}$ values in Eq. (10) should be transposed with $R_{rtm}$, assuming the spectral dependency term $k$ in Eq. (2) holds constant within the wavelength pair set. Then the pre-constructed LUT of $R$ values can be utilized to examine the sensitivity of $R$ values to errors in input parameters and can be used in estimating the uncertainty range of the $NO_2$ VCD. Furthermore, we can transpose the ratio between $R_{rtm,A}$ and $R_{rtm,B}$ as $Q$, and reformulating Eq. (10) as the
univariate equation of $Q$ (Eq. 11). This transposition is necessary to avoid the necessity examining the correlation between $R_{rtm,A}$ and $R_{rtm,B}$ for the error propagation.

$$VCD_{True} = \frac{a_B \cdot (b_A - b_B)}{a_B - (\frac{R_{obs,A}}{R_{obs,B}}) \cdot a_A} + b_B = \frac{a_B \cdot (b_A - b_B)}{a_B - (\frac{R_{rtm,A}}{R_{rtm,B}}) \cdot a_A} + b_B = \frac{a_B \cdot (b_A - b_B)}{a_B - Q \cdot a_A} + b_B \qquad (11)$$

Under these premises, the error of $VCD_{True}$ can be estimated from Eq. (11) as follows (Eq. 12).

$$\sigma^2_{VCD_{True}} = \left( \frac{a_A \cdot a_B \cdot (b_A - b_B)}{(a_B - Q \cdot a_A)^2} \right) \cdot \sigma^2_Q \qquad (12)$$

The uncertainty of $Q$ value ($\sigma_Q$) is determined by the sensitivity of $Q$ values to the uncertainty of input parameters. However,
$\sigma_Q$ has a dependency on $Q$ values, so the uncertainty of the normalized $Q$ value ($\sigma_{Q_{rel}}$), which better portrays the relative uncertainty, has been calculated from Eq. (13) under the assumption that all the parameters are uncorrelated. It is noteworthy to mention that the $\sigma_{Q_{rel}}$ can be easily converted to $\sigma_Q$ by multiplying the $Q$ value calculated from the HIS spectrum in the actual $NO_2$ VCD retrieval.

$$\sigma^2_{Q_{rel}} = \Sigma_{var} \left( \left| \frac{\partial Q_{norm}}{\partial var} \right|^2 \cdot \sigma^2_{var} \right) \qquad (13)$$



Table 4 presents the assumed uncertainty range of input parameters ($\sigma_{var}$) used in estimating $\sigma_{Q_{rel}}$, and Fig. 3 (also, see supplementary Figs. S4 and S5) is an example of the sensitivity test results showing $\frac{\partial Q_{norm}}{\partial var}\sigma_{var}$ values under the reference condition (presented in supplementary Table S2). Most of the uncertainties were posed by the uncertainties in the
calibrations, with the greatest contribution from the uncertainties of spectral SF calibration followed by the uncertainties in spectral shift calibration. The uncertainty of the spectral SF calibration largely depended on the quality of the clean pixel used for the spectral SF derivation. For instance, when the spectral SF is calculated with clean pixel data obtained over a homogeneous surface and with the nearly steady-signal state, the uncertainty propagated to the $Q$ value is minimal. The general relative uncertainty of a spectral SF in converting the raw HIS signal to the non-uniformity corrected spectra was
kept below $10^{-2}$ and $10^{-1}$ for the clean pixel over the ocean and the land surface, respectively. The accuracy of the spectral calibration was generally high, with the spectral shift uncertainty constrained within 0.05 nm on most of the occasions. However, $Q$ values were even sensitive to the small spectral shifts, resulting in the second largest contributor to the total uncertainty.

The relative significance of the instrument noise level is characterized by SNR and is dependent on the signal intensity
measured by the HIS. Assuming a SNR of $2\times10^3$, which is a typical value in between the scenes over the bright land surface and the dark ocean surface, yielded the instrument noise to become a succeeding source of error.

Among the variables composing the LUT for the MWP method, surface reflectance and PBLH were the most influential parameters in terms of uncertainties. PBLH from ERA5 reanalysis data (Hersbach et al., 2023b) was adopted for an application of the MWP method because there was only a subset of research flights having explicit observation upon PBLH,
such as from lidar, located at the vicinity of the target domain. The PBLH from lidar observations and the ERA5 reanalysis were comparable for the research flights with the lidar observation nearby (not shown), but the uncertainty of region-specific PBLH in the global reanalysis dataset was regarded conservatively considering that the target domains are mostly located near the coastline where the PBLH can exhibit significant spatial variability.

The Suomi-NPP VIIRS BRDF data (VNP43C1; Schaaf et al., 2019) was used to calculate effective surface reflectivity
accounting for the viewing geometry during the airborne HIS observation. Moreover, HIS observations with extreme viewing zenith angle were discarded in the retrieval, leaving only a small space for an error in surface reflectance. However, the spatial inhomogeneity of surface reflectance can be significant; hence the uncertainty of the surface reflectance should be considered. The sensitivity of $Q$ values to the surface reflectance was greater at the darker surface, probably attributed to the perturbation amplified in terms of relative change at low-reflectivity conditions due to a smaller denominator value. It is
pertinent to mention that the uncertainties in $Q$ values, induced by the uncertainties in PBLH and surface reflectance, become greater under high $NO_2$ VCD conditions. $NO_2$ becomes more concentrated at the low altitudes as the column density increases, coupled with the fact that both PBLH and surface reflectivity control the scattering weights of the low altitudes, resulting in a greater sensitivity of the $Q$ values to the PBLH and surface reflectivity perturbations under high $NO_2$ VCD condition.





Solar zenith angle and the observing altitudes are almost certain, considering the accuracy of chronographs and GPS sensor used for the research flights. Nonetheless, the time synchronization between the instruments might have a small discrepancy, and the narrow window of uncertainty has been accounted for in the sensitivity test. Furthermore, the viewing zenith angle and the relative azimuth angle can show moderate uncertainty from the misalignment between the HIS and the IMU sensor module, whereas both variables showed minimal impact on the $Q$ value uncertainty.

The LUT of the sensitivities upon uncertainties of each input parameter ($\frac{\partial Q_{norm}}{\partial var} \sigma_{var}$) has been constructed for every set of wavelength pairs, incorporating the conditions that the $R$ value LUT spans, for an uncertainty estimation of NO$_2$ VCDs retrieved from the MWP method.

**Table 4. Uncertainty range of variables ($\sigma_{var}$) assumed for the error estimation.**

| Variables | Uncertainty ($\sigma_{var}$) | Unit | Remarks |
|---|---|---|---|
| PBLH | 0.4 | km | - |
| Albedo (Reflectivity) | 0.03 | - | - |
| Solar Zenith Angle | 0.05 | Degree (°) | - |
| Flight Altitude | 10 | m | - |
| Viewing Zenith Angle | 3 | Degree (°) | - |
| Relative Azimuth Angle | 3 | Degree (°) | Uncertainty mainly attributed to VAA |
| Wavelength shift | 0.05 | nm | Wavelength shift calibration fitting uncertainty |
| Instrumental noise | Variable | - | Depending on the observed signal intensity |
| Radiometric Calibration | Variable | - | Depending on which spectral SF applied for the retrieval |








**Figure 3. Sensitivity of Q values calculated from simulated radiances at wavelength-pair 1 (i.e., Type_A: 414.209, 415.535 nm; Type_B: 417.126, 418.452 nm) depending on (a) NO₂ VCD, (b) PBLH, (c) albedo (ALB; reflectivity), (d) solar zenith angle (SZA), (e) observation altitude (ALT), (f) viewing zenith angle (VZA), and (g) relative azimuth angle (RAA) considering atmospheric condition at Pohang and its corresponding reference conditions (shown in supplementary Table S2).**




## 4 Airborne observations of NO₂ VCDs

### 4.1 Research flights

Three regions in Korea were selected as the target domain for the research flight: Chungnam, Jecheon, and Pohang. Each domain has one or more industrial point sources with substantial NO₂ emissions (Choo et al., 2023; Kim et al., 2020;
Cleansys, 2023), such as power plants, steel yards, petrochemical complexes, and cement kilns. Figure 4 depicts the composite TROPOMI NO₂ VCD data (version 2.0.4; Copernicus Sentinel data processed by ESA, 2021) spanning four years (from 2019 to 2022) for October and November, which aligns with the timeframe of the research flights, with the locations of the target domains as well as the precise position of the point sources within each domain. TROPOMI NO₂ VCDs generally displayed relatively low concentrations compared to the year-round average (Park et al., 2022), with the highest
concentration observed at the southeastern side of the Seoul Metropolitan Area (SMA; 0.70 DU). Although the greatest peak is found over the SMA, noticeable increases in NO₂ VCDs were observed in all three target domains. The NO₂ peaks become even more conspicuous and localized near the major industrial sources as the TROPOMI composite is downscaled to a finer resolution (Fig. 4), providing solid evidence of significant NO₂ emissions from these specific point sources.

The details of the research flights for airborne HIS observation and the configurations of the HIS mounted on the airplane are
shown in supplementary Table S3 and Fig. S6. A total of five research flights were conducted, with three flights extensively dedicated to the Chungnam domain. The HIS was mounted on gyro-stabilized mounts in the Cessna 208 aircraft, whereas an external canister was used to mount the HIS on the pylon beneath the Beechcraft 1900D aircraft. The gyro-stabilized mount is expected to have dampened the vibrations for the research flights with the Cessna airplane, while the HIS mounted on an external pylon might have been exposed to a moderate level of vibrations during the flights with the Beechcraft 1900D. The
position and observation geometry during the airborne HIS observation were obtained from IMU/GPS module dedicated to the HIS in principle, but the airplane's IMU/GPS information was used for the research flights with Beechcraft 1900D after calibration upon the difference between HIS viewing geometry and the airplane's plane of reference.

The actual flight path and altitude of each research flight are illustrated in supplementary Fig. S7. Flight altitude was limited to 11,000 ft due to air traffic regulations in Korea and unpressurized aircraft cabin. The low flight altitude, combined with
the narrow FOV of the HIS (approximately 13°), resulted in a narrow observation swath ranging between 340 m (at 5,000 ft) and 750 m (at 11,000 ft).





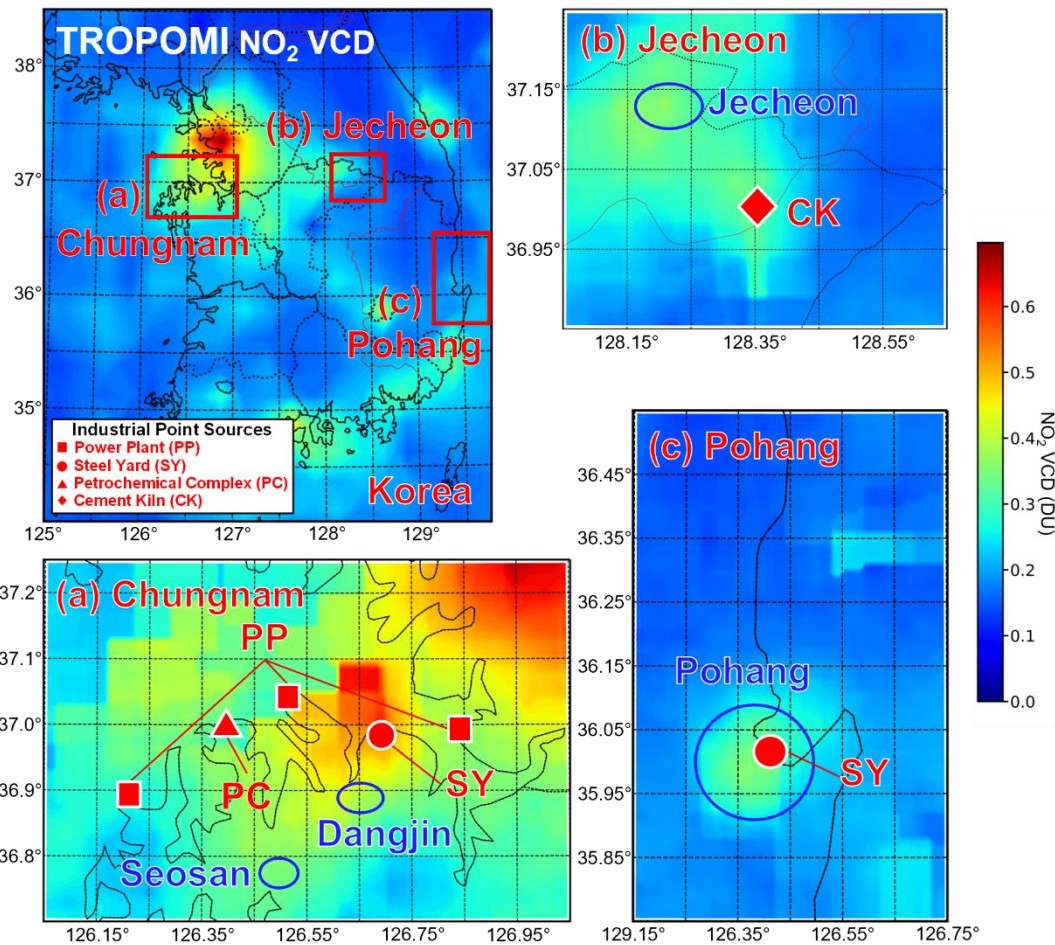

**Figure 4. Location of major industrial point sources in Korea targeted for airborne HIS observations overlayed by average**
**TROPOMI NO₂ VCD_Total in October and November from 2019 to 2022. Corresponding TROPOMI swath datasets were downscaled to 0.25°×0.25° and 0.01°×0.01° resolutions for Korea and each respective target domain (i.e., Chungnam, Jecheon, and Pohang, respectively).**

## 4.2 NO₂ VCDs in three industrial areas

The MWP method applied to all three sets of wavelength pairs yields three independent NO₂ VCD with corresponding uncertainty estimates from a single HIS spectrum. To take full advantage of three independent NO₂ VCDs, the weighted mean value was taken as a single representative NO₂ VCD value for the HIS spectrum from a specific spatial column and frame. The weights were determined to be inversely proportional to the squared uncertainty of each NO₂ VCD from three different wavelength pair sets (Eq. 14), which can minimize the random uncertainty theoretically (Xie et al., 2001).






$$VCD_{final} = \sum_{i=1}^{3}(\alpha_i VCD_i), \quad where \ \alpha_i = \frac{(\frac{1}{\sigma_i^2})}{\sum_j(\frac{1}{\sigma_j^2})} \tag{14}$$

The narrow swath of the airborne HIS observation resulted in a footprint of pre-binned spatial columns to have an extremely short across-track length scale (approximately 27 m at 11,000 ft and 13 m at 5,000 ft) compared to the along-track length

scale (approximately 100 m under aircraft ground speed of 360 km h$^{-1}$ and a 1 second of exposure). Moreover, the remaining uncertainties of the NO$_2$ VCDs from the pre-binned spatial columns were still excessive, thus the additional post-binning was conducted. To be specific, the weighted average was taken over NO$_2$ VCDs from all 50 pre-binned spatial columns and four consecutive frames (integrated exposure time of 4 seconds) with consideration of uncertainties of each NO$_2$ VCD (as shown in Eq. 14). Therefore, the final HIS data has a footprint almost symmetric at an altitude of 6,000 ft (400 m × 400 m),

but can be variable according to the ground speed of an airplane and the flying altitude.

The NO$_2$ VCDs from the airborne HIS observations and their uncertainties are shown in Figs. 5, 6, and 7, together with the TROPOMI swath data of the same day. The NO$_2$ VCDs observed from three research flights over the Chungnam area showed different spatial patterns due to different wind fields (Fig. 5). The highest NO$_2$ concentration was observed on 24 November 2022 when the wind was calm (< 2 m s$^{-1}$) within the boundary layer (Fig. 5e). Since the atmosphere was stagnant

near the surface, the peaks of NO$_2$ VCDs were observed right above the major emission point sources. The highest concentration was found over the steel yard, with the peak HIS NO$_2$ VCD reaching up to 2.0 DU, while the same peak from the TROPOMI swath showed 1.0 DU. Likewise, the HIS captured peaks with higher NO$_2$ VCDs over the petrochemical complex (> 1.2 DU), while the NO$_2$ VCD of the corresponding TROPOMI footprint remained relatively low (0.9 DU). While the small elevated NO$_2$ VCD signals were discernible over some of the power plants from the airborne HIS

observations, TROPOMI failed to capture those minor peaks.

The research flights on 17 October 2020 (Fig. 5c) and 25 November 2022 (Fig. 5g) exhibited relatively low NO$_2$ VCDs than the flight on 24 November 2022, attributed to relatively strong surface wind rapidly dissipating the emitted plumes from the point sources. For instance, Fig. 5g shows a clear depiction of NO$_2$ plume conically dispersed and transported to the north under clear southerly wind at the boundary layer. The significance of NO$_2$ plume emitted from a steel yard was also

emphasized in Fig. 5g, whereas the stronger wind spreading out the plume into a wider area resulted in lower NO$_2$ VCDs compared to the previous day (Fig. 5e). Meanwhile, the observation area of the research flight on 17 October 2020 (Fig. 5c) was relatively narrow compared to other flights at Chungnam, only incorporating the petrochemical complex as a major emission point source within the HIS observation area. The wind direction was between northerly and northwesterly, with moderate wind speed at the surface, and the NO$_2$ plume emitted from the petrochemical complex was clearly distinguished

with a dispersing pattern toward the southeast.

It is worth pointing out that in Fig. 5g, the TROPOMI shows elevated NO$_2$ VCDs on the eastern side of the plume detected by the HIS, originating from a steel yard. The average wind field at the area above the boundary layer was strong westerly



wind, which can be presumed from faster ground speed bounding east (approximately 370 km h$^{-1}$) compared to the ground speed of the west-bound flights (approximately 290 km h$^{-1}$). Therefore, it can be inferred that the NO$_2$ plumes below the

aircraft at an altitude of 6,000 ft tended to propagate northward in alignment with the surface wind field, whereas the plumes at higher altitudes were advected east. This emphasizes the importance of considering the three-dimensional space to elaborate the dispersion and transport pathways of plumes emitted from point sources, as well as accounting for the wind fields at different altitudes. By doing so, the differences between the collocated passive measurements with different viewing geometries, such as satellite, airborne, and ground-based observations, can be better understood.






**Figure 5. (a)** Downscaled average TROPOMI NO₂ VCD in October and November from 2019 to 2022 and **(b)** Google Earth image (© Google Earth) of the Chungnam domain with major industrial point sources of NO₂ and populated urban areas denoted in red and blue circles, respectively. **(c), (e),** and **(g)** show NO₂ VCD retrieval results from HIS overlayed on TROPOMI swath data and **(d), (f),** and **(h)** show the uncertainty of HIS-retrieved NO₂ VCD. **(c)** and **(d), (e)** and **(f),** and **(g)** and **(h)** show observed results from the research flight on 17 October 2020, 24 November 2022, and 25 November 2022, respectively.





The NO₂ VCDs were slightly elevated over the Jecheon area on 3 November 2020, compared to the average NO₂ VCDs of October and November from 2019 to 2022 (Fig. 6). The NO₂ plumes transported from the SMA (located approximately 100 km to the west) is likely responsible for the general increase of NO₂ VCDs throughout the area, since the strong westerlies

prevailed both within the boundary layer and the free troposphere. However, the effect of the plume was limited since the strong wind must have dispersed the SMA plume in wide area lowering the concentration. The uncertainties of the HIS NO₂ VCDs were exceptionally high at Jecheon, with the absolute uncertainty around 0.3 DU, almost 100% in relative uncertainty. This can be explained by combined effect of the higher absolute uncertainty level due to low-quality clean pixel used for a SF correction, and generally low NO₂ VCD leading to greater relative uncertainty per same absolute uncertainty. Still, both

TROPOMI and HIS were able to capture some increased NO₂ VCDs right above the cement kiln, which has been pointed as a major industrial point source of NO₂ at the area (Kim et al., 2020).

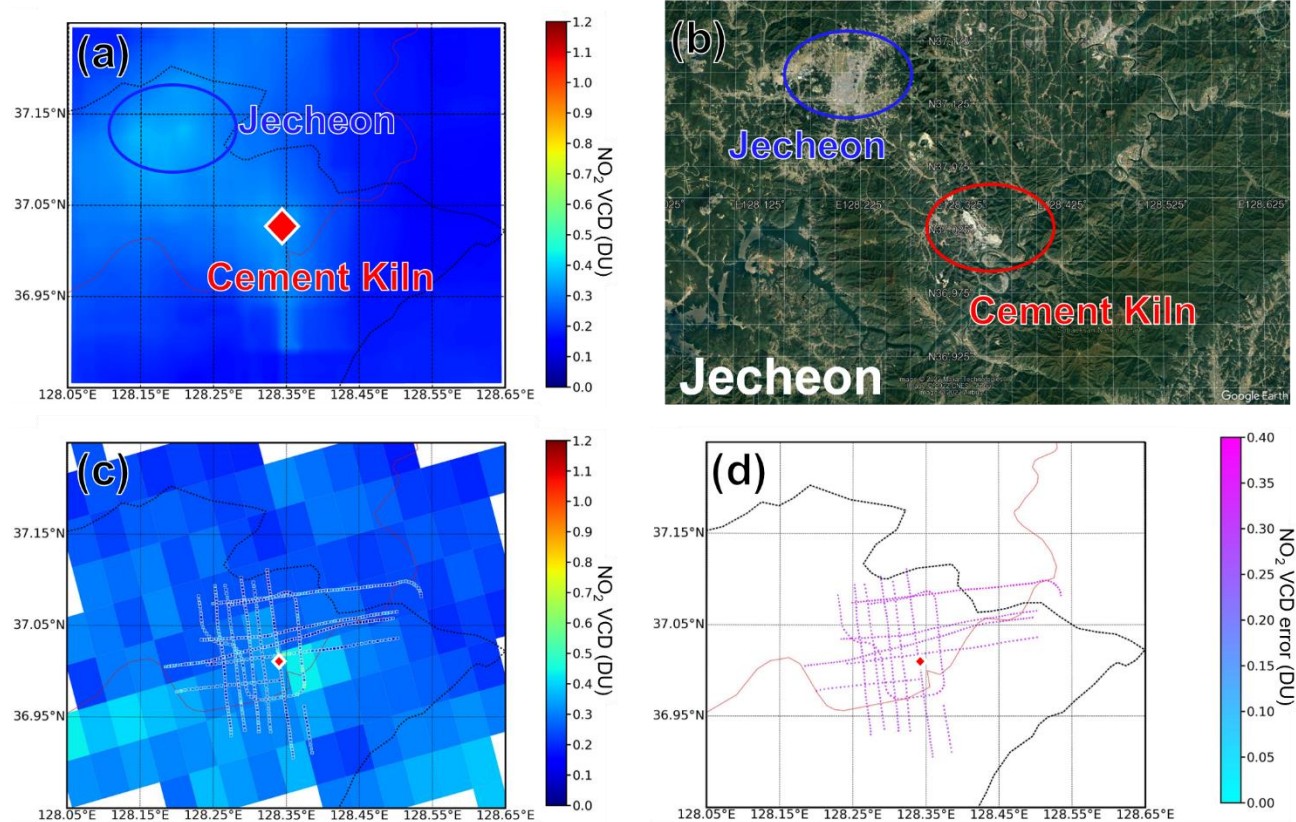

**Figure 6. (a) Downscaled average TROPOMI NO₂ VCD in October and November from 2019 to 2022 and (b) Google Earth image**
**(© Google Earth) of the Jecheon domain with major industrial point source of NO₂ and populated urban area denoted in red and blue circles, respectively. (c) and (d) show observed results from the research flight on 3 November 2020, while (c) shows NO₂ VCD retrieval results from HIS overlayed on TROPOMI swath data, and (d) shows the uncertainty of HIS-retrieved NO₂ VCD.**



The research flight over the Pohang area further demonstrated the significance of $NO_2$ emissions from the steel yards, since

one of the world's largest steel yard is located within the city of Pohang (Fig. 7). Southerlies were observed near the surface, and the $NO_2$ VCDs were higher at the northern side of the steel yard. Since the city of Pohang is much populated than the cities shown before (i.e., Jecheon, Seosan, Dangjin), the $NO_2$ VCDs even from the HIS showed peak over a wider spatial range. Nevertheless, the highest $NO_2$ VCDs were observed right above the steel yard with a peak concentration of 1.8 DU, which is a way beyond the values that can be observed from satellites (i.e., TROPOMI) at the region. For instance, the

highest TROPOMI $NO_2$ VCD was around 0.9 DU. The TROPOMI pixel with the highest VCD was located right next to the exact pixel corresponding to the location of the steel yard, and on the northern side. This conforms with southerly-dominant atmospheric condition at the time of observation, but is shifted toward north compared to those found in the airborne HIS observation. The temporal disparities between the HIS frames collected near the steel yard and the TROPOMI overpass were insignificant (< 30 minutes; shown in supplementary Fig. S11). Therefore, as an extent to the discussion upon Fig. 5g, the

conical dispersion and transport of a plume in three-dimensional space must have caused the spatial shift of the peaks from a different viewing geometry of satellite and the airborne sensors.





**Figure 7. (a) Downscaled average TROPOMI NO₂ VCD in October and November from 2019 to 2022 and (b) Google Earth image**
**(© Google Earth) of the Pohang domain with an industrial point source of NO₂ and populated urban area denoted in red and**
**white circles, respectively. (c) and (d) show observed results from the research flight on 5 November 2020, while (c) shows NO₂**
**VCD retrieval results from HIS overlayed on TROPOMI swath data, and (d) shows the uncertainty of HIS-retrieved NO₂ VCD.**

The NO₂ VCDs from the HIS and the TROPOMI generally showed good agreement, considering the maximum time difference of 2.5 hours between the HIS observation and the TROPOMI overpass, as well as the substantial temporal variability of NO₂ in the vicinity of the emission sources (Park et al., 2022). The correlation coefficient (R) ranged from 0.59 to 0.73 (supplementary Figs. S9 and S11), except for the research flights at Jecheon with relatively low correlation (R=0.4; supplementary Fig. S10). The exceptionally low correlation of the HIS and TROPOMI NO₂ VCDs at Jecheon can be explained by the distinctively high uncertainty of the HIS NO₂ VCDs.



Meanwhile, the HIS NO$_2$ VCDs exhibited higher levels of uncertainty over the ocean surface (0.10–0.15 DU) than the land surface (0.025–0.075 DU), when the uncertainties associated with the SF calibrations were ruled out. The higher uncertainty observed over the ocean surface can be attributed to the lower surface reflectivity of the ocean (as depicted in supplementary Fig. S8). The lower reflectivity can lead to a reduction in the SNR, resulting in an increased level of uncertainty arising from instrumental noise. Furthermore, it can also contribute by itself as an increasing factor of retrieval uncertainty, as discussed

in Sect. 3.4 (shown by Fig. 3 and supplementary Figs. S4 and S5).

## 4.3 Satellite sub-grid variability of NO$_2$ near the industrial point sources

The NO$_2$ VCDs from the airborne HIS observations exhibited greater variability than the collocated TROPOMI. In other words, the HIS have shown considerable sub-grid variability within a footprint of TROPOMI. To elaborate the difference between the HIS and TROPOMI in respect of satellite sub-grid variability, HIS and TROPOMI NO$_2$ VCDs from all five

research flights were collected. Before comparing the collected set of collocated HIS and TROPOMI NO$_2$ VCDs, bias offsets were incorporated into the HIS NO$_2$ VCDs per flight (more precisely, per spectral SF). This adjustment aimed to ensure that the mean bias between the HIS and the TROPOMI NO$_2$ VCD to be zero for a dataset using the same spectral SF, because the HIS NO$_2$ VCDs can show systematic biases depending on which reference NO$_2$ VCD was used to calculate spectral SF. Therefore, it is necessary to remove the systematic biases incurred from the spectral SF correction to avoid an unintended

increase of mean absolute error (MAE) when all the datasets from different flights are combined for a comparison.

As a result, Fig. 8a shows a scatter plot comparing the bias-corrected HIS NO$_2$ VCDs and the collocated TROPOMI data, collected from all five research flights conducted for this study. Since the bias-corrected dataset was used for comparison, the mean bias was zero. The overall correlation coefficient (R) was 0.73, which was almost equivalent to the highest correlation found in individual flights. The MAE, which can be the indication of the average TROPOMI sub-grid variability

incorporating all the research flights, was 0.106 DU. However, considering the random uncertainty of the HIS NO$_2$ VCD ranging from 0.025 DU (bright land surface) to 0.15 DU (dark ocean surface) under premises that the adequate calibrations were applied (i.e., clean pixel correction), these random uncertainties posed in the HIS NO$_2$ VCDs can explain most of the parts of the mean absolute error. Accordingly, it may be inappropriate to argue that the mean absolute error of 0.106 DU represents the overall sub-grid variability at the targeted domains.

Figure 8b is a more explicit depiction of the TROPOMI sub-grid variability according to its NO$_2$ VCDs. From the quantile values of the HIS NO$_2$ VCDs calculated for every 0.1 DU bin of the TROPOMI NO$_2$ VCDs, the differences between the quantiles (maximum – minimum to 51$^{st}$ – 49$^{th}$ quantiles) according to TROPOMI NO$_2$ VCD bins are shown. An evident inclination has been identified on the spread of the HIS NO$_2$ VCD quantile values as the collocated TROPOMI NO$_2$ VCD increases. In specific, the difference between 75$^{th}$ and 25$^{th}$ HIS NO$_2$ VCD quantile values for the TROPOMI footprints with

NO$_2$ VCD below or equal to 0.6 DU yielded 0.15 DU on average, while for the TROPOMI footprints with NO$_2$ VCD greater than or equal to 0.8 DU yielded 0.52 DU.





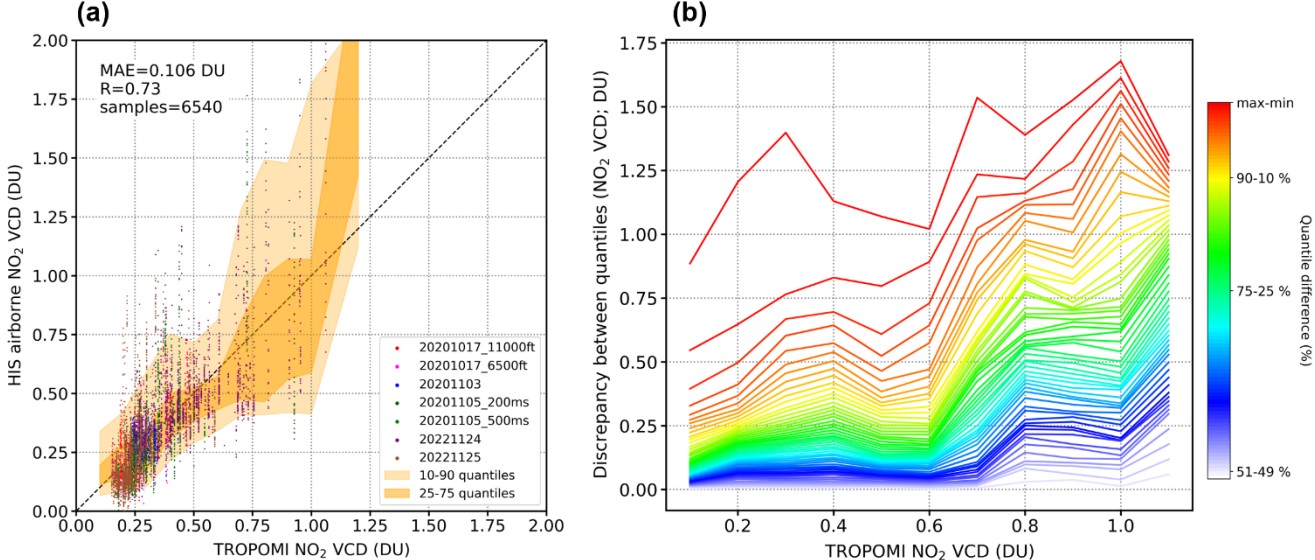

**Figure 8. (a) Comparison of bias-corrected HIS and TROPOMI NO₂ VCDs with the corresponding HIS NO₂ VCD quantiles (10, 25, 75, 90-th) shown over the TROPOMI NO₂ VCD range. (b) Differences between HIS NO₂ VCDs quantile values (from maximum-minimum to 51%-49%) at corresponding TROPOMI NO₂ VCD range.**

Various artifacts, even besides the retrieval uncertainty, contribute to the variabilities expressed in Fig. 8. Hence, it is challenging to isolate the actual contribution of satellite sub-grid variability. For instance, temporal discrepancies between the HIS frames and the TROPOMI overpass, as well as the combined effect of different viewing geometry and the complex vertical structure of the wind fields, can result in meaningful disparities between the two collocated measurements. Nevertheless, the notably greater HIS NO₂ VCD variability at TROPOMI footprints with high NO₂ VCDs compared to footprints with relatively low NO₂ VCD suggests a noticeable increase in satellite sub-grid variability near significant emission sources. As an extent, the inherent spatial resolution limitations of the satellites can impact their capability to monitor plumes in close proximity to emission sources. To gain a better understanding of the atmospheric conditions near emission hot spots, it is essential to utilize complementary high-resolution observations, such as airborne hyperspectral observations.

# 5 Summary and conclusions

In this study, we developed the versatile NO₂ VCD retrieval algorithm, so-called the Modified Wavelength Pair (MWP) method, and have applied it to the spectra obtained from the airborne low-cost hyperspectral imaging sensor (HIS) observations at three (Chungnam, Jecheon, and Pohang) industrial areas in Korea. The MWP method is based on the pseudo-linear relationship between the wavelength pair radiance ratio ($R$) and the NO₂ VCD. The Look-Up Table (LUT) of $R$ values



for each wavelength pair was constructed using forward radiative transfer simulations with UVSPEC, which encompassed all possible conditions of the airborne HIS observations. An analytical solution has been derived to calculate $NO_2$ VCD

using the $R$ values calculated from the airborne observed spectrum and the corresponding $R$ values in the LUT. This analytical solution allows for the retrieval of three independent $NO_2$ VCD values from a single HIS spectrum, one for each set of wavelength pairs. The merits of the MWP method are as follows: 1) Applicable to the spectra obtained from instruments with relatively low spectral resolution and volatile characteristics, attributed to the lack of precise maintenance or adequate temperature–vibration management, 2) Insensitive to broadband spectral tendencies such as from aerosol and

surface reflectivity, and 3) Computationally competitive from the simplified LUT calculations. For instance, high-resolution chemical transport model simulations are not essentially required.

An analytical derivation of the "True" $NO_2$ VCD in the MWP method enables feasible error estimation based on sensitivity tests upon the LUT. The greatest source of uncertainty was the uncertainty in spectral scale factor (SF) calibration, which largely depended on the quality of the clean pixel. The uncertainty in the spectral shift calibrations and the instrument noise

was the succeeding source of uncertainty, whereas the uncertainties in variables constructing the LUT were relatively minor. The final airborne HIS $NO_2$ VCD data at a spatial resolution of approximately 400 m × 400 m (at a flying altitude of 6,000 ft) exhibited uncertainty ranging from 0.025 DU (for bright land surfaces) to 0.15 DU (for dark ocean surfaces), except for the research flight at Jecheon where only a low-quality clean pixel was applicable.

The $NO_2$ plumes emitted from steel yards, both in Chungnam and Pohang, were particularly prominent among the point

sources, while stagnant atmospheric conditions with low surface wind speed were a prerequisite for elevated $NO_2$ concentrations at the proximal area to the emission sources. The overall correlation coefficient ($R$) between the collocated HIS and TROPOMI $NO_2$ VCDs was 0.73, with a mean absolute error of 0.106 DU. The correlation between the two was limited by temporal disparities between the HIS frames and the TROPOMI overpass, as well as the different observation geometries combined with complex vertical wind fields. There was an apparent increase in the variability of HIS $NO_2$ VCDs

corresponding to the TROPOMI footprints as the TROPOMI $NO_2$ VCD increased. This implies an amplification of satellite sub-grid variability in $NO_2$ VCDs near the center of the plume. Therefore, the inherent limitations of satellite spatial resolution can significantly impact its utility in monitoring plumes near emission sources, and it is necessary to employ complementary high-resolution observations. When low-cost, feasible, and compact hyperspectral imagers, such as the HIS used in this study, are utilized to acquire high-spatial resolution observations, it will lead to an improved understanding of

atmospheric conditions near emission point sources through more feasible and frequent observations.

**Acknowledgments** This study was funded by the Fine Particle Research Initiative in East Asia Considering National Differences (FRIEND) through the National Research Foundation of Korea (NRF), funded by the Ministry of Science and ICT (Grant No.: 2020M3G1A1114615) and NIER research grant (NIER-SP2022-268). Jin-Soo Park, Hyun-Jae Kim, and

Jinsoo Choi were supported by NIER research grant (NIER--2023-01-01--142).



**Code and Data availability** The airborne HIS NO$_2$ VCD data are publicly available at https://doi.org/10.7910/DVN/YCZ9JU (Park, 2023). The TROPOMI NO$_2$ VCD datasets are available at NASA GES DISC (https://disc.gsfc.nasa.gov/datasets/S5P_L2__NO2____HiR_2/summary?keywords=tropomi; Copernicus Sentinel data processed by ESA, 2021). The ERA5 reanalysis datasets are available at https://doi.org/10.24381/cds.bd0915c6 (Hersbach et al., 2023a) and https://doi.org/10.24381/cds.adbb2d47 (Hersbach et al., 2023b). VIIRS BRDF datasets are from https://lpdaac.usgs.gov/products/vnp43c1v001/ (Schaaf et al., 2019). Other datasets such as the raw airborne HIS spectra, the Look-Up Table, ancillary datasets for calibration, and associated codes for their processing may be shared on request to the corresponding author.

**Competing interests** The authors declare that they have no conflict of interest.

**Author contributions** Jong-Uk Park: Conceptualization, Formal analysis, Methodology, Visualization, Writing - original draft. Hyun-Jae Kim, Jin-Soo Park, and Jinsoo Choi: Data curation, Funding acquisition, Project administration. Sang Seo Park: Conceptualization, Methodology, Writing - review & editing. Kangho Bae, Jong-Jae Lee, and Chang-Keun Song: Data curation, Writing - review, and editing. Soojin Park, Kyuseok Shim, and Yeonsoo Cho: Data curation, Writing – review and editing. Sang-Woo Kim: Conceptualization, Funding acquisition, Writing - review and editing, Supervision.

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
