# Peer review of "Airborne observation with a low-cost hyperspectral instrument: Retrieval of NO2 VCD and the satellite sub-grid variability over industrial point sources"

_EGUsphere, 2023_

## Author Comment (AC1)

**Response to Reviewers' Comments**

Dear Editor and Reviewer:

We appreciate for your thoughtful and helpful comments. We tried to answer your comments, and our 'Response' is embedded below. All changes we have made in the revised manuscript have not only been mentioned in each response for the reviewer's comments, but also marked with an MS Word tracking option. We hope we have provided the appropriate answers and corresponding modifications. If there are any further questions, please let us know.

Best Regards,

Authors

**Authors' response to RC2 from referee # 1:**

**General Comments**

This paper describes a simplified retrieval for NO$_2$ from solar backscatter measurements, based on wavelength pair ratios (on/off absorption spectral lines)—the Modified Wavelength Pair (MWP) method—designed for use with low-cost hyperspectral sensors that lack the measurement stability of satellite and more-expensive airborne instruments. This technique is applied to the Hyperspectral Imagining Sensor (HIS), which was flown on aircraft over three significant pollution sources in Korea. An analytical uncertainty analysis is included. Results are compared with TROPOMI retrievals, and sub-satellite-grid-scale differences are discussed, in the context of geophysical variations.

The manuscript is well written, thorough, and generally clear. I recommend minor revisions.

**Specific Comments**

Paragraph starting in L48: Mention TEMPO and other, geostationary spacecraft, which are also achieving fairly good spatial resolution information.

→ Thank you for your suggestion. However, the paragraph is intended to describe the development of Low-Earth Orbit (LEO) satellites and to emphasize that the S5P (Sentinel-5 Precursor) TROPOMI, which has been used in this study for comparison with the airborne HIS measurements, has significantly enhanced spatial resolution than its predecessors, but is still struggling to resolve the spatial inhomogeneity of trace gases such as $NO_2$. Including additional descriptions about the recent geostationary satellite instruments such as the Geostationary Environment Monitoring Spectrometer (GEMS; Kim et al., 2020) or Tropospheric Emissions: Monitoring of Pollution (TEMPO; Chance et al., 2013; Zoogman et al., 2017) might be more informative, but none of them are currently producing official public-release datasets yet. Moreover, we believe that not mentioning the geostationary satellites in the paragraph would be helpful in terms of the overall fluency of the introduction; hence, we decided to keep it as it was in the original manuscript.

References

Chance, K. V., Liu, X., Suleiman, R. M., Flittner, D. E., Al-Saadi, J. and Janz, S. J.: Tropospheric emissions: monitoring of pollution (TEMPO), In Proc. SPIE 8866, Earth Observing Systems XVIII, doi:10.1117/12.2024479. http://dx.doi.org/10.1117/12.2024479, 2013.

Kim, J., Jeong, U., Ahn, M. H., Kim, J. H., Park, R. J., Lee, H., Song, C. H., Choi, Y. S., Lee, K. H., Yoo, J. M., Jeong, M. J., Park, S. K., Lee, K. M., Song, C. K., Kim, S. W., Kim, Y. J., Kim, S. W., Kim, M., Go, S., Liu, X., Chance, K., Miller, C. C., Al-Saadi, J., Veihelmann, B., Bhartia, P. K., Torres, O., Abad, G. G., Haffner, D. P., Ko, D. H., Lee, S. H., Woo, J. H., Chong, H., Park, S. S., Nicks, D., Choi, W. J., Moon, K. J., Cho, A., Yoon, J., Kim, S. kyun, Hong, H., Lee, K., Lee, H., Lee, S., Choi, M., Veefkind, P., Levelt, P. F., Edwards, D. P., Kang, M., Eo, M., Bak, J., Baek, K., Kwon, H. A., Yang, J., Park, J., Han, K. M., Kim, B. R., Shin, H. W., Choi, H., Lee, E., Chong, J., Cha, Y., Koo, J. H., Irie, H., Hayashida, S., Kasai, Y., Kanaya, Y., Liu, C., Lin, J., Crawford, J. H., Carmichael, G. R., Newchurch, M. J., Lefer, B. L., Herman, J. R., Swap, R. J., Lau, A.

K. H., Kurosu, T. P., Jaross, G., Ahlers, B., Dobber, M., McElroy, C. T. and Choi, Y.: New era of air quality monitoring from space: Geostationary environment monitoring spectrometer (GEMS), B. Am. Meteorol. Soc., 101, E1–E22, https://doi.org/10.1175/BAMS-D-18-0013.1, 2020.

Zoogman, P., Liu, X., Suleiman, R., Pennington, W., Flittner, D., Al-Saadi, J., Hilton, B., Nicks, D., Newchurch, M., Carr, J., Janz, S., Andraschko, M., Arola, A., Baker, B., Canova, B., Miller, C. C., Cohen, R., Davis, J., Dussault, M., Edwards, D., Fishman, J., Ghulam, A., Abad, G. G., Grutter, M., Herman, J., Houck, J., Jacob, D., Joiner, J., Kerridge, B., Kim, J., Krotkov, N., Lamsal, L., Li, C., Lindfors, A., Martin, R., McElroy, C., McLinden, C., Natraj, V., Neil, D., Nowlan, C., O'Sullivan, E., Palmer, P., Pierce, R., Pippin, M., Saiz-Lopez, A., Spurr, R., Szykman, J., Torres, O., Veefkind, J., Veihelmann, B., Wang, H., Wang, J., and Chance, K.: Tropospheric emissions: Monitoring of pollution (TEMPO), J. Quant. Spectrosc. Ra., 186, 17–39, https://doi.org/10.1016/j.jqsrt.2016.05.008, 2017.

Table 1: Please include some information about the f-number, etendue, and/or SNR of the system (under a given set of circumstances), that would indicate the optical throughput of the system.

→ We have added some additional information (i.e., f-number) of the HIS in Table 1 of the revised manuscript. However, due to our unawareness regarding the optical hardware, we decided not to add the information we are not entirely confident about. Further details about the instruments can be found in "HIS_Headwall_specifications.pdf", which has been added in a supplementary material repository at https://doi.org/10.7910/DVN/YCZ9JU (please note that the HIS has an ADC depth of 16 bits, not 14 bits as stated in the document). Furthermore, discussions about the SNR of the system have been discussed in section 3.3, and no further thorough descriptions are available from the manufacturing perspective.

Section 3.1: Please reference the use of wavelength pairs in other retrievals. For example, wavelength pairs were long used in the retrieval of total ozone.

→ There have been previous studies that used one or more wavelength pairs to retrieve columnar concentrations of atmospheric $O_3$ and $NO_2$, which are gases with significant

optical depth in the atmosphere in the UV–VIS band attributed to their abundance and strong absorption feature. These studies utilized ground-based instruments such as the Dobson spectrophotometer (Dobson, 1957a; 1957b) or the Brewer spectrophotometer (Brewer, 1973; Brewer and Kerr, 1973; Brewer et al., 1973). We appreciate your suggestion and have added some sentences and corresponding references in lines 130–134 of the revised manuscript.

References

Brewer, A. W.: A Replacement for the Dobson Spectrophotometer?, Pure and Applied Geophysics (PAGEOPH), 106–108, 1973.

Brewer, A. W. and Kerr, J. B.: Total Ozone Measurements in Cloudy Weather, Pure and Applied Geophysics (PAGEOPH), 106–108, 1973.

Brewer, A. W., McElroy, C. T., and Kerr, J. B.: Nitrogen Dioxide Concentrations in the Atmosphere, Nature, 246, 1973.

Dobson, G. M. B.: Observers' handbook for the ozone spectrophotometer, in Annals of the International Geophysical Year, V, Part 1, 46-89, Pergamon Press, 1957a.

Dobson, G. M. B.: Adjustment and calibration of the ozone spectrophotometer, ibid. V, Part I, 90-113, Pergamon Press, 1957b.

Section 3.1: By combining pairs of wavelengths, one with the shorter wavelength having the stronger absorption ("Type A") and the other with longer wavelength having the stronger absorption ("Type B"), you in effect partially cancel the bias from spectrally changing surface reflectivity (since the forward model does not include reflectivity spectral dependence). I suggest you say that explicitly. It would make the mathematical discussion easier to understand.

→ Thank you for the suggestion. The exact reason why we used a set of wavelength pairs comprised of one Type_A wavelength pair and one Type_B wavelength pair, instead of using a randomly matched dual-wavelength pair, was to minimize the effect of random uncertainties on the final analytical solution shown in Eq. (10). When the same type of wavelength pair (i.e., Type_A or Type_B) is used in Eq. (10), both the numerator and the

denominator of the first term on the right-hand side are likely to be excessively small in absolute values resulting in amplified uncertainties. Therefore, it is necessary to ensure that $a_A$ and $a_B$ have different signs (as well as $b_A$ and $b_B$) to ensure sufficiently large values for the denominator and numerator in Eq. (10). We tried to avoid stating too many technical details in the manuscript, which we believe is unnecessary and somewhat veiling the results of the airborne HIS $NO_2$ VCD observations. Instead, we have added a brief explanatory sentence in line 179 of the revised manuscript as an alternative.

Throughout the entire manuscript, I recommend using r (small r) for the correlation coefficient, to reduce confusion with the radiance ratio R (capital R).

→ Thank you for the suggestion. We tried to differentiate the radiance ratio $R$ with the correlation coefficient R throughout the manuscript, but we agree that following your advice would be helpful for readers to avoid unnecessary confusion. Therefore, we have modified accordingly throughout the revised manuscript.

Paragraphs starting in L182: This section is confusing. In L183, what does "k-fold of the value" mean? The definition of k spectral dependency factor—is not introduced until L195. R is then discussed—I think meaning the radiance ratio—but since reflectivity is also being discussed, it is tempting to think R means reflectivity. In L184, "The same k value can be assumed for the wavelength pairs..."—why can that be assumed? In L193, "relation between the biased VCD estimates"—biased in what way? Why are they biased? Do you mean they include measurement errors?

→ The whole paragraph you are referring to is meant to describe the situation where the spectral dependency of real-world reflectivity differing from assumed model-world reflectivity (i.e., RTM-input surface reflectivity). We assumed arbitrary unspecified constant $k$ to represent the difference in spectral dependency of real-world and model-world reflectivity, and this difference contributes to the discrepancy of model-world radiance ratio ($R_{rtm}$) and real-world radiance ratio ($R_{obs}$) as shown in Eq. (2). We thought we first stated "$k$" by definition in line 183 of the original manuscript, while the statement in line 195 ("spectral dependency factor $k$") was a simple repetition for the reminder. We never used a symbol or the acronym for reflectivity in Sect. 3.1. and further assured that

the "$R$" only stands for radiance ratio throughout the manuscript by following your advice to change the symbol of correlation coefficients to "r".

The grounds for assuming the same "$k$" value for the wavelength pairs in their vicinity (i.e., < 5 nm) is because what spectral dependency term $k$ represents is the effect of smoothly varying terms with small high-spectral-frequency variabilities such as surface or aerosol reflectivity. Please refer to the explanations in lines 192–196 of the revised manuscript.

We referred $VCD_{obs}$ as a biased VCD estimate because it is an estimation based on the HIS-observed radiance ratio value ($R_{obs}$) and the regression coefficients (i.e., $a, b$ in Eq. 1) retrieved from the relation between simulated radiance ratio ($R_{rtm}$) and input NO₂ VCD to the RTM (Eq. 3). The aforementioned $k$, or the discrepancy of spectral dependency of reflectivity between the model and the real world (i.e., observations), causes the HIS-observed radiance ratio value ($R_{obs}$) to differ from the anticipated radiance ratio value ($R_{rtm}$). What we should retrieve from the radiance ratio affected by the unknown spectral dependency of reflectivity in the real world (i.e., $R_{obs}$) is the $VCD_{true}$, or the RTM-input VCD that resulted in the $R_{rtm}$ and the corresponding regression coefficients (i.e., $a, b$ in Eq. 1) representing the $R - VCD$ relationship.

L202, Eq (4): Equating VCD_True with VCD_rtm,A and VCD_rtm,B is confusing. If the modeled VCDs were "true," observations wouldn't be needed. Maybe I don't understand what "True" means in this context?

→ (As an extent to our response on the previous question) The concept of Eq. (4) is that regardless of which radiance ratio was used, the resulting VCD should be the same in principle. The whole point of the MWP method is to cancel out the spectral dependency term of reflectivity so that the NO₂ VCDs can be retrieved from the observed radiance ratios (i.e., $R_{obs,A}$, $R_{obs,B}$) and the VCD-R relation identified from the RTM simulations. $VCD_{True}$ is a value that we would like to retrieve from a set of observed radiance ratios, not affected by the spectral dependency term "$k$".

L232: "three independent NO₂ VCD"—I understand what you mean, but they aren't really "independent." "Different" may be more accurate.

→ Thank you for the advice, and we have changed it accordingly in the revised manuscript (line241).

L234: "to increase the signal-to-noise ratio" and L236: "spectral binning of ±2 original spectral bins"—It would be good to capture this in Table 1, so the reader doesn't look at the table and this and think you have 5x spectral oversampling. The table would be more useful if it reflected the sampling/binning and SNR that are used in the retrieval.

→ We agree that the phrase "Spectral binning interval" in Table 1 can confuse readers that the spectra we used are effectively 5 times oversampled, which is not true. We appreciate your comment pointing this out, and have modified the corresponding row of Table 1 in the revised manuscript to clarify. The way we have estimated the SNR is from the dark observations; thus, our SNR estimates vary from scene to scene depending on the signal intensity. Therefore, we were unable to provide certain SNR value in a table. However, we have accounted for the SNR per each spectrum considering all the pre-binning (i.e., spectral and spatial binning) while estimating the uncertainties presented in Figs. 5~7.

L254: "highly resolved CTM data": Highly resolved in what way? Horizontal spatial? Vertically? Also, I would not say models produce "data"—they produce "output." Measurements are data.

→ Thank you for the comment. We have modified the corresponding sentence in line 264 of the revised manuscript to avoid referring to model output as "data", and further clarified that the high resolution refers "spatial" resolution. Enhancement of CTM vertical resolution will also help us to obtain a more realistic $NO_2$ vertical profile, but studies show that the (horizontal) spatial resolution has a greater influence on the accuracy of CTM outputs (Kim et al., 2016; Valin et al., 2011; Zhao et al., 2020).

References

Kim, H. C., Lee, P., Judd, L., Pan, L., and Lefer, B.: OMI $NO_2$ Column Densities over North American Urban Cities: The Effect of Satellite Footprint Resolution, Geoscientific Model Development, 9 (3), 1111–1123, doi:10.5194/gmd-9-1111-2016, 2016.

Valin, L. C., Russell, A. R., Hudman, R. C., and Cohen, R. C.: Effects of Model Resolution on the Interpretation of Satellite $NO_2$ Observations, Atmospheric Chemistry and Physics, 11 (22), 11647–11655, doi:10.5194/acp-11-11647-2011, 2011.

Zhao, X., Griffin, D., Fioletov, V., McLinden, C., Cede, A., Tiefengraber, M., Müller, M., Bognar, K., Strong, K., Boersma, F., Eskes, H., Davies, J., Ogyu, A., and Lee, S. C.: Assessment of the Quality of TROPOMI High-Spatial-Resolution $NO_2$ Data Products in the Greater Toronto Area, Atmospheric Measurement Techniques, 13 (4), 2131–2159, doi:10.5194/amt-13-2131-2020, 2020.

Paragraph staring in L366: Q is not clearly defined. What does "transpose the ratio between R_rtm,A and R_rtm,B as Q" (L369) mean? No Q ppears in Eq. (11).

→ The sentence "transpose the ratio between $R_{rtm,A}$ and $R_{obs,A}$ as $Q$" means that the $Q$ is defined as the $R_{rtm,A}$ over $R_{rtm,B}$ (i.e., $Q = \frac{R_{rtm,A}}{R_{rtm,B}}$). $Q$ appears in the denominator of the right-hand side of Eq. (11).

L472: "mean value"—How well did the three different VCDs agree? It would be useful to know how similar they are, and if they're different, why.

→ Theoretically, all three VCD values from three different wavelength pair sets should agree well. However, there are cases where three VCD values do not agree well in the actual retrieval. To minimize the unintended biases by simply taking the "mean value" of all three VCD values, we have taken the weighted averaging technique proposed by Xie et al. (2001) to minimize the uncertainty of the "weighted mean" value from the given data points and their respective uncertainties. Therefore, we would like to claim that the "weighted mean value" of the three VCDs, considering their respective uncertainties, has more or less reduced the uncertainty to a certain extent on every occasion, rather than deteriorating the uncertainty range.

Reference

Xie, S. X., Liao, D., and Chinchilli, V. M.: Measurement error reduction using weighted average method for repeated measurements from heterogeneous instruments, Environmetrics, 12(8), 785–790. https://doi.org/10.1002/env.511, 2001.

L484: Footprint is 400 m x 400 m, so this is essentially the entire swath width, correct? It would be helpful to state that.

→ Yes, indeed it is. The entire across-track pixels were post-binned to a single pixel, therefore yielding single-pixel HIS swath data with approximately 400 m swath width (at a flying altitude of 6,000 ft). However, the swath width varied by the observation altitude, becoming wider under higher altitudes and vice versa. We have added an explicit statement emphasizing that the entire across-track pixels were binned to a single post-binned across-track pixel as a final HIS data (please refer to lines 505–506 in the revised manuscript).

Figures 4/5/6/7: How were the TROPOMI data downscaled?

→ Figures 4, 5a, 6a, and 7a are the visualization of downscaled TROPOMI $NO_2$ $VCD_{Total}$ (entitled "$VCD_{summed}$" in TROPOMI product) composites, from TROPOMI swaths in October and November from 2019 to 2022. First, the downscaled grid has been pre-determined for each figure domain (i.e., 0.25°×0.25° for Korea, 0.1°×0.1° for the rest of the domains). For every downscaled grid point and TROPOMI swath, TROPOMI pixels located within approximately ± 2 pixel width from the downscaled grid point were selected, and the distances between the downscaled grid point and the selected TROPOMI pixels' center were calculated. Using the inverse of those distances as a weight, the weighted average was taken as a $NO_2$ VCD value for a single downscaled grid point in a typical TROPOMI swath. Finally, the average of the $NO_2$ VCD values corresponding to each downscaled grid point were taken and has been visualized as shown in the figures. It is noteworthy that all TROPOMI $NO_2$ VCD swath data were screened with quality flags (qa_value > 0.75).

The details of the downscaling methodology (as shown above) were omitted in the manuscript because those figures (i.e., Figs. 4, 5a, 6a, and 7a) were intended to show general $NO_2$ VCD distribution in each respective target domain at a similar time of year,

without elaborating with detailed discussions. The figures you have pointed out may appear peculiar, attributed to the limited availability of TROPOMI swath data over the target domains in October and November. We now perceive that some readers might find figures puzzling. Consequently, we have added a few more details on the revised manuscript (Fig. 4 caption).

L575: "Before comparing the collected set of collocated HIS and TROPOMI $NO_2$ VCDs, bias offsets were incorporated into the HIS $NO_2$ VCDs"—How well do HIS and TROPOMI agree in general, before bias are removed? I am interested in inherent retrieval bias as well as the representativeness of the TROPOMI footprint. In the Abstract and Summary, it is stated that 0.106 DU is the absolute error of the measurement, in comparison to TROPOMI. Is that true? It seems like some bias has already been removed.

→ The $NO_2$ VCDs retrieved from the airborne HIS observations and using the MWP method have inherent limitations in estimating the absolute magnitude of $NO_2$ VCD. To be more specific, in the spectral SF calibration using the clean pixel data, we need to estimate the $NO_2$ VCD of the clean pixel from external data sources (i.e., CTM, satellite, etc.). That is why we tried to select the clean pixels in the upwind region of the major NOx sources to minimize the discrepancy or the bias attributed to CTM simulated outputs or the satellite (i.e., TROPOMI) data. Nevertheless, the "existing" bias between assumed $NO_2$ VCD in the clean pixel to the real-world $NO_2$ VCD at the particular time and scene directly poses a bias to all of the HIS $NO_2$ VCDs calibrated with particular spectral SF (or the clean pixel data). To minimize this bias, we have further adjusted the $NO_2$ VCD assumed for the clean pixel for comparison with the TROPOMI data.

After all these efforts to minimize an "universal" bias relevant to the clean pixel $NO_2$ VCD assumption, we still found that HIS $NO_2$ VCDs and TROPOMI showing mean bias in different signs by flight-to-flight. We aimed to show the increasing intra-pixel $NO_2$ VCD variability according to the satellite $NO_2$ VCD increment in Fig. 8, whereas simply combining all the HIS data containing the mean bias attributed from the clean pixel $NO_2$ VCD assumption will increase the overall MAE (Mean Absolute Error) or the RMSE (Root Mean Square Error). Therefore, we have removed the mean bias of the HIS $NO_2$ VCD compared to the collocated TROPOMI data per research flight.

The MAE value of 0.106 DU presented in the abstract and the summary is the value representing the spread of the HIS data compared to the TROPOMI data after removing the mean bias. The bias removal would indeed have reduced the MAE by its value. However, we presented the MAE value not to emphasize the accuracy of HIS-driven $NO_2$ VCDs, but to show the increasing intra-pixel variability within a satellite pixel in the vicinity of industrial point sources. We are aware that the MAE value, or what Fig. 8 exhibits, does not essentially show the sole effect of the satellite intra-pixel variabilities, and we have tried to explain it to a certain extent considering the possible other uncontrolled factors (i.e., retrieval uncertainties) affecting the HIS-TROPOMI intercomparison.

**Technical Corrections**

The English usage is generally good. I include several suggestions that I noted when reviewing the paper, along with technical comments, below.

→ Thank you for your valuable advice and suggestions. We sincerely appreciate your comments and have made modifications based on your recommendations, unless specifically mentioned otherwise in the comments below.

Line 12:

"The high spatial"—remove "The"

"(VCDs) were measured"—replace "measured" with "retrieved" (VCDs are not measured, strictly speaking)

"from the airborne"—remove "the"

→ We have changed them in our revised manuscript (line 12).

L24: "The typical"—remove "The"

→ We have changed in our revised manuscript (line 24).

L29: "different observation geometries under complex vertical wind fields"—The winds don't change the geometry, per se. Better to say something like "different pollution distributions..."

→ Our intention was to highlight that differences in the observation geometries of satellite and airborne instruments can lead to discrepancies in the spatial distribution of retrieved NO$_2$ VCD, consequently limiting the correlation between spatially collocated satellite and airborne observations. For instance, when the horizontal wind field varies below and above the flying altitude of the aircraft, the plume will be advected in different directions below and above the aircraft according to the respective wind field, yielding different NO$_2$ VCD distribution patterns observed from the satellite and the aircraft. We have made slight modifications on the revised manuscript to make this statement more straightforward (lines 28–29)

L37: "pollutant"—change to "pollutants"

→ We have changed in our revised manuscript (line 38).

L65:

"calibrations"—change to "calibration"

"retain"—change to "maintain"

→ We have changed in our revised manuscript (line 66).

L75: "East Asia, where the"—remove "the"

→ We have changed in our revised manuscript (line 76).

L78: "calibrations"—change to "calibration"

→ We have changed in our revised manuscript (line 79).

L91: "latest (spectral rows)"—change to "spectral rows at the edges"

→ We have changed in our revised manuscript (line 93).

L92: "grating is used with a concave mirror"—Is it a concave grating, or is there a mirror in addition? Please clarify. Also, no where do you say if HIS uses reflective or transmissive optics. I am assuming reflective.

→ Yes, HIS uses reflective optics as you presume, and has a concave mirror in addition to the diffraction grating. The details about the instrument are available in "HIS_Headwall_specifications.pdf", which has been added in a supplementary material repository at https://doi.org/10.7910/DVN/YCZ9JU.

L100: "Unlike the"—change to "Despite these"

→ We have changed in our revised manuscript (line 101).

L122: "DOAS fitting"—change to "the DOAS fitting"

→ We have changed in our revised manuscript (line 123).

L127: "compartments"—change to "components"

→ We have changed in our revised manuscript (line 129).

L152: "following the convention"—remove; this is confusing and unnecessary

→ We have changed in our revised manuscript (line 159).

L164: "condition"—change to "conditions"

→ We have changed in our revised manuscript (Fig. 2 caption).

L175: three dots before "VCD" (mathematical symbol meaning "because")—remove; not necessary

→ We have changed in our revised manuscript (Eq. 1).

L240: "The UVSPEC"—remove "The"

→ We have changed in our revised manuscript (line 250).

Table 3 headings: "Unit"—change to "Units"

→ We have changed in our revised manuscript (Table 3 caption).

Table 3 Solar Zenith Angle and Flight Altitude: Are variable, but would you report the range used? (the column has the other ranges)

→ The SZA and ALT entries for the LUT were adjusted for each research flight to avoid redundant computation, considering the possible range of SZA (i.e., calculated based on local time, latitude, and day of the year) and recorded range of ALT (supplementary Fig. S7). Therefore, SZA and ALT entries, as well as their range, vary by flight-to-flight. We thought elaborating on the details of LUT entries could be unnecessary, but we acknowledge your comment and have tried to add some valuable remarks in Table 3 of the revised manuscript. Explanations of the additional statements that have been added to Table 3 are as follows:

SZA: SZA entries were in 3° intervals for all the flights, and ranged from 42° (for the research flight at 17 October 2020) to 75° (for the research flight at 5 November 2020 and 24 November 2022; observations under SZA > 70° were filtered out for the retrieval). Number of SZA entries varied by flight-to-flight mainly attributed to the total duration of each research flight.

ALT: The number of ALT entries varied from 4 (on 3 November 2020, 5 November 2020, and 25 November 2022) to 9 (on 24 November 2022), primarily influenced by the number of cruising altitudes during each research flight. For instance, research flights with only four ALT entries were those conducted at a single predetermined altitude

without any issues during the flight. On the other hand, cruising altitude changed during the flight on 24 November 2022 due to an unexpected order from the air traffic control, resulting in the enlarged number of ALT entries. We tried to make sure ALT entries were in 100 m intervals for ± 100 m range from the cruising altitude, while some additional ALT entries were considered in between the multiple cruising altitudes (i.e., if the cruising altitudes of the research flight were 1.5 km and 3.4 km a.m.s.l., ALT entries were determined as [1.4, 1.5, 1.6, 3.3, 3.4, 3.5 + $\alpha$] km a.m.s.l.).

L253: "legitimate"—change to "realistic"

→ We have changed in our revised manuscript (line 263).

L258: "23 vertical grid"—23 layers over what altitude/pressure range? 70 hPa top noted in L274, but it should be stated here. Same comment for L266.

→ For the CMAQ model, 23 vertical layers were determined from the terrain height (surface) to 70 hPa, whereas 31 vertical layers for WRF model (apologies that the original manuscript was misleading–made the correction in the revised manuscript line 276) ranged from surface to 50 hPa with hydrostatic pressure coordinates. Moreover, MCIP (Meteorology-Chemistry Interface Processor) module was used to apply WRF-simulated outputs to CMAQ. We have made modifications according to your comment and added some descriptions regarding the vertical range of CMAQ/WRF models in the revised manuscript (lines 268–269 and 276).

L292: "inferring"—change to "implying"

→ We have changed in our revised manuscript (line 303).

L315: "should be calibrated"—change to "was calibrated"

→ We have changed in our revised manuscript (line 326).

L329: "clean pixel"—define what this means when it's first used (here)

→ The description of "clean pixel" has been expatiated on the following paragraph (please refer to lines 346–349 in the revised manuscript), and we believe that the current narrative structure is efficiently conveying the detailed description of the "clean pixel". We kindly ask for your consent to keep the corresponding paragraphs as it was in the original manuscript.

L375: "premises"—change to "circumstances"

→ We have changed in our revised manuscript (line 386).

L400: "Assuming a SNR"—change to "Assuming an SNR"

→ We have changed in our revised manuscript (line 411).

L586: "premises—change to "circumstances"

→ We have changed in our revised manuscript (line 613).

L609: "As an extent"—do you mean "To a certain extent"?

→ What we intended was closer to such expressions as "accordingly" or "moreover", rather than "to a certain extent". Therefore, we have changed the phrase in our revised manuscript (line 637).

L623: "volatile"—change to "unstable"

→ We have changed in our revised manuscript (line 651).

L630: "succeeding"—change to "principal"

→ We intend to express that the uncertainties in the spectral shift calibrations and the instrument noise were a second-major group contributing to the total uncertainties, succeeding the primary source of uncertainty, the uncertainty in spectral scale factor calibration. Therefore, we believe that the word "succeeding" better represents our

intention than the "principal"; hence, we decided to keep the sentence as it was in the original manuscript.

**Again, we appreciate for your valuable comments and suggestions.**

===End of Document===

---

## Author Comment (AC2)

**Response to Reviewer's Comments**

Dear Editor and Reviewer:

We appreciate for your thoughtful and helpful comments. We tried to answer your comments, and our 'Response' is embedded below. All changes we have made in the revised manuscript have not only been mentioned in each response for the reviewer's comments, but also marked with an MS Word tracking option. We hope we have provided the appropriate answers and corresponding modifications. If there are any further questions, please let us know.

Best Regards,

Authors

**Authors' response to RC1 from referee # 2:**

**Summary**

In their work, Park et al. motivate and present a new method to derive $NO_2$ VCDs from low cost sensors that can be used on airplanes without regular maintenance. The Modified Wavelength Pair (MWP) method uses measured ratios at two wavelengths in combination with radiative transfer simulations to estimate the $NO_2$ VCD. While the method has some limitations in terms of precision and accuracy, the approach with low cost and low maintenance is a good addition to ground based or satellite borne measurements.

The paper is generally well written, the scientific approach is motivated and well described. The presentation of the results and satellite comparisons leave some room for improvement. Therefore, some revisions and technical corrections are listed below.

**General/Major revisions**

As the Hyperspectral Image Sensor (HIS) is used specifically for the new Modified Wavelength Pair approach, it should be compared to other low-cost sensors. For example: How does the method compare to $NO_2$ camera (Dekemper et al., 2016) in precision, detection limit and versatility?

→ Thank you for your comment and your suggestion to review a research paper about $NO_2$ column density retrieval using a method other than the DOAS fitting (i.e., Dekemper et al., 2016). We found that the AOTF-based $NO_2$ camera described in Dekemper et al. (2016) is more suited as a ground-based stationary instrument, targeting on observing direct stack plumes rather than aiming to obtain raster images from airborne observations. The AOTF-based $NO_2$ camera employs a similar approach to the MWP method in this study, utilizing radiances from a wavelength pair composed of one wavelength with strong sensitivity and another with weak sensitivity to the $NO_2$ absorption. However, the $NO_2$ camera can only observe radiance at a single wavelength at a time, hence, sequential measurements in different wavelengths are required for the retrieval. This hampers the utility of $NO_2$ camera in airborne observations since the aircraft cannot remain stationary over a specific location.

While the $NO_2$ camera can capture a 2-D spatial structure in a single snapshot, making it advantageous for monitoring the cross-sectional behavior of stack plumes, it can only measure the optical depth difference between strong- and weak-absorbing wavelengths and, therefore, the $NO_2$ slant column density (SCD) along the line of sight. Furthermore, selecting wavelengths in close proximity (i.e., a few nanometers apart) can be enough to exclude the effect from broad-band spectral characteristics induced by aerosols in ground-based sky observations. However, during airborne downward observations, the spectral features from both the surface and aerosols intermingle and can significantly affect radiances in two different wavelengths, even when those wavelengths are closely spaced.

Therefore, push-broom hyperspectral instruments with grating systems and 2-D detectors (i.e., CCD) cannot be replaced by such instruments as $NO_2$ camera for airborne observations. In this study, we focus on versatility in terms of airborne $NO_2$ VCD observations. Therefore, although the principle of $NO_2$ column density retrieval from the

NO$_2$ camera is interesting, we have decided not to delve into such discussions in detail in the introduction.

Reference

Dekemper, E., Vanhamel, J., Van Opstal, B., and Fussen, D.: The AOTF-based NO$_2$ camera, Atmos. Meas. Tech., 9, 6025–6034, https://doi.org/10.5194/amt-9-6025-2016, 2016.

With a wavelength resolution of 1.4nm (FWHM) it should be possible to perform a DOAS analysis. Comparing a classical DOAS approach to the newly developed MWP method would help to understand advantages and applications of both approaches. This should at least be included in the supplement.

→ The HIS used in this study should have 1.4 nm FWHM according to the specification provided by the manufacturer (Headwall Photonics, Inc.; please refer to the instrument specification document entitled "HIS_Headwall_specifications.pdf", which has been added in a supplementary material repository at https://doi.org/10.7910/DVN/YCZ9JU), which can be sufficient enough for applying DOAS approach to retrieve differential slant column densities (dSCDs). Unfortunately, the actual optical characteristics of the HIS were not only different from those claimed by the manufacturer but also variable. For instance, the spectral registration and the slit function FWHM retrieved from monochromatic emission lines of the Hg-lamp (Ocean Optics, Inc.; HG-2 model) showed significant variation by frame-to-frame and also along the spatial and spectral axis of the CCD (Figs. R1 and R2).

[Figure]

Figure R1. The HIS slit function FWHM retrieved with the reference Hg-lamp assuming asymmetric super-Gaussian slit function. Exposure time was set to 100 ms, with no saturated pixel. The "Hg line number" in the y-axis refers to Hg-lamp emission lines at 296.728, 313.155, 365.015, 404.656, and 435.833 nm, respectively.

[Figure]

Figure R2. The frame-to-frame variation (i.e., standard deviation) of the HIS spectral registration retrieved from the reference Hg-lamp observations. A total 97 frames with an exposure time of 100 ms were included. Wavelengths were registered to spectral pixels by quadratic spline interpolation (or extrapolation).

The limited signal-to-noise ratio (SNR) was another major obstacle in applying both the DOAS-fitting and the MWP method. Therefore, spectral binning (i.e., ± 2 spectral pixels binned) was applied prior to applying the MWP method, whereas such spectral binning complicates the simultaneous fitting of the slit function FWHM in the DOAS fitting. Moreover, spatial binning on the HIS spectra prior to DOAS-fitting is inadequate, considering the large across-pixel variability of spectral registration and slit function FWHM.

The only viable option to utilize the DOAS-fitting technique is to apply DOAS fitting to every spectrum measured from the original across-track pixels, while considering a wide range of slit function FWHM. This requires excessive computational resources and is vulnerable to convergence failure due to the wide range of FHWM given, associated with low SNR. With regard to all the limitations of the HIS optical characteristics, applying an alternative approach for reasonable $NO_2$ VCD retrieval was necessary.

Nevertheless, we have tried to apply DOAS fitting using the QDOAS software (Danckaert et al., 2017), considering the fitting window of 410–450 nm as shown in Figs. R3 ~ R5. To minimize the number of fitting variables for the calibration (i.e., slit function FWHM), we used the wavelength-shift-calibrated HIS spectra as an input to the QDOAS software. However, slit function FWHM retrieved from the fitting referencing to the high-resolution extraterrestrial solar irradiance spectrum (i.e., SAO2010) still exhibited excessive across-track variability (see Figs. R4 and R5; results from the nearby across-track pixel showing very different FWHM). This is a concerning result, casting doubt on the reliability of DOAS-fitted results. Moreover, as shown in Figs. R3~R5, fitting residuals were significant, making the fitted results (i.e., $NO_2$ dSCDs) meaningless. Therefore, we found it challenging to show nor compare the results with the conventionally retrieved DOAS-based $NO_2$ VCDs. We agree that it would be fruitful to compare $NO_2$ VCDs from the DOAS-based approach and the MWP method with airborne observations using delicate instruments such as GeoTASO, but we believe it is beyond the scope of our study.

[Figure]

Figure R3. The DOAS-fitting results with the wavelength shift corrected spectra measured during the airborne HIS observations on 25 November 2022 (2nd across-track pixel out of 139). (a) Calibration results from fitting with the high-resolution solar irradiance spectrum, and (b) DOAS fitted spectra weighted by fitted value for each parameter (red) overlayed by the residuals added (black). Synthetic reference spectrum calculated from the spectra measured over the clean pixel was used for the DOAS fitting, whereas "SFP1" in (a) refers to slit function FWHM assuming Gaussian distribution.

[Figure]

Figure R4. Same as Fig. R3, but the results from 81st across-track pixel out of 139.

[Figure]

Figure R5. Same as Fig. R3, but the results from 82nd across-track pixel out of 139.

We admit that we have skipped the detailed discussion about the shortcomings of the MWP method and would like to discuss them here in the response. Compared to the conventional DOAS-fitting-based approach, the MWP method relies on the partial information contained in the spectrum. Therefore, the MWP inherently has a broader range of certainty (i.e., greater uncertainty) than the DOAS-fitting-based retrievals. This can be an acceptable "minor" downside of the MWP method in case of $NO_2$ retrievals in the early–mid 400 nm bands. However, the limited use of spectral information can fundamentally preclude the retrieval of other atmospheric gaseous constituents with their major absorption band coinciding with multiple absorbing gases (i.e., $SO_2$, HCHO, etc.). Furthermore, we have traded off the spectral information and the sensitivity to constrain the effects from excessive noise and instrumental instabilities by applying the moving average to the spectra prior to applying the MWP method. Bearing the limitations of the MWP method, we agree that it would be better to apply conventional DOAS-fitting in case of airborne UV-VIS hyperspectral observations with stable instruments.

Reference

Danckaert, T., Fayt, C., and Van Roozendael, M.: QDOAS Software user Manual, Royal Belgian Institute for Space Aeronomy, https://uv-vis.aeronomie.be/software/QDOAS/index.php, 2017.

Application/comparison: While the comparison to TROPOMI looks really nice (especially in the easier-to-read figures S9 and S10), it would be really interesting to see a comparison to ground based DOAS instruments – if any are available. TROPOMI is known two underestimate the $NO_2$ VCD in heavily polluted regions and thus a comparison to ground-based instruments (maybe during GMAP/SIJAQ) would make it easier to evaluate the results of the MWP approach.

→ We agree that, in addition to the comparison with TROPOMI satellite, the comparison with $NO_2$ VCDs measured from the collocated fiducial ground-based measurements (i.e., Pandora) would add greater value to this study and can be an explicit evaluation of $NO_2$ VCD retrievals with the MWP method. Unfortunately, we have no flights above the exact location of Pandora (or any other ground-based DOAS instruments) since the Pandora

observations in Korea were mainly focused on major megacities like Seoul and Busan, where airborne observations are prohibited due to airport traffic and military restrictions. Nevertheless, the Pandora network in Korea has recently increased, and more Pandora instruments are and will start observation. Therefore, we are expecting flying over ground-based Pandora sites to become much more feasible. Thank you again for your constructive comment, and we will plan flight tracks considering the Pandora site locations in our future airborne HIS observations.

Figures 3 to 8 are generally difficult to understand. A detailed list of suggested/required improvements for each Figure will be attached at the end of this review.

→ Thank you for pointing out some points in our drawings. We have answered each respective comment below in detail.

The used TROPOMI data product needs some description as to how gridding for the comparison to the MWP data results was done. Details like cloud filtering need to be considered as they could strongly influence the $NO_2$ VCD results (e.g. Boersma et al., 2004, Eskes et al. 2020).

→ We conducted the research flights considering the weather and cloud coverage because our primary goal was to measure $NO_2$ VCDs with the HIS. Therefore, most research flights were conducted in regions with cloud-free to cloud-less conditions. However, some flights were conducted under cloudy conditions despite our vigilance regarding the weather forecast. These flights have been excluded from this paper, and we believe that the TROPOMI swath data collocated with airborne HIS data selected for the research would have minimal complication due to clouds.

Nevertheless, we have checked the TROPOMI data used in the analysis in terms of cloud fraction (i.e., effective cloud area fraction assuming fixed cloud albedo retrieved from the $NO_2$ spectral window entitled "cloud_fraction_crb_nitrogendioxide_window" or "$f_{eff, NO_2}$"; Eskes et al., 2022; 2023, van Geffen et al., 2022) and the maximum $f_{eff, NO_2}$ of the TROPOMI pixel data incorporated in the analysis was 0.166 (on November 24, 2022). Accordingly, every $NO_2$ $VCD_{Total}$ TROPOMI swath data used in the analysis was quality

assured and had "qa_value" greater than 0.75, which is the recommended criteria suggested in the TROPOMI NO$_2$ Product User Manual (Eskes et al., 2022).

There was no additional gridding applied to the TROPOMI L2 swath data for the analysis (except for Figs. 4, 5a, 6a, and 7a; details about those figures are elaborated in response to the question below), and HIS NO$_2$ VCD data points within the TROPOMI pixel were intercompared with each other (i.e., Fig. 8 and supplementary Figs. S9 ~ S11 in revised manuscript). Namely, multiple HIS data points can be collocated with a single TROPOMI pixel.

We have added some descriptions on the revised manuscript regarding the TROPOMI NO$_2$ data used in this study (line 459) together with the general weather conditions (i.e., cloud conditions) during the research flights (lines 469–470) to be more descriptive.

References

Eskes, H. J., Eichmann, K.-U., Lambert, J.-C., Loyola, D., Stein-Zweers, D., Dehn, A., and Zehner, C.: S5P Mission Performance Centre Nitrogen Dioxide [L2__NO2__] Readme, V.2.4 (S5P-MPC-KNMI-PRF-NO2), 2023.

Eskes, H. J., van Geffen, J., Boersma, F., Eichmann, K.-U., Apituley, A., Pedergnana, M., Sneep, M., Veefkind, J. P., and Loyola, D.: Sentinel-5 precursor/TROPOMI Level 2 Product User Manual Nitrogen Dioxide, V.4.1.0 (S5P-KNMI-L2-0021-MA), 2022.

van Geffen, J. H. G. M., Eskes, H. J., Boersma, K. F., and Veefkind, J. P.: TROPOMI ATBD of the total and tropospheric NO$_2$ data products, V.2.4.0 (S5P-KNMI-L2-0005-RP), 2023.

**Minor revisions**

Line 90: "is known to be non-linear when the CCD counts exceed approximately 80% of the saturation level" – do you have a source for this claim?

→ It may be impetuous to say that all CCD detectors share the same non-linearity character. Meanwhile, it is well-known that CCD gain alters as it reaches near the saturation level (i.e., 75 ~ 80 % of the maximum counts), and additional calibration is

required to use the data acquired near the saturation level. We apologize for missing the references after the statement and have added references in the revised manuscript (lines 91–92).

References

Nehir, M., Frank, C., Aßmann, S., Achterberg, E.P.: Improving Optical Measurements: Non-Linearity Compensation of Compact Charge-Coupled Device (CCD) Spectrometers, Sensors, 19 (12), 2833, https://doi.org/10.3390/s19122833, 2019.

Pulli, T., Nevas, S., El Gawhary, O., van den Berg, S., Askola, J., Kärhä, P., Manoocheri, F., and Ikonen, E.: Nonlinearity characterization of array spectroradiometers for the solar UV measurements, *Appl. Opt.*, 56, 3077-3086, https://doi.org/10.1364/AO.56.003077, 2017.

Wang, S., Carpenter, D. A., DeJager, A., DiBella, J. A., Doran, J. E., Fabinski, R. P., Garland, A., Johnson, J. A., and Yaniga, R.: A 47 million pixel high-performance interline CCD image sensor, *IEEE Trans. Electron Devices,* 63 (1), 174–181, https://doi.org/10.1109/TED.2015.2447214, 2016.

Eqn. (3): Maybe switch the last term with the second to last term to get a fully logical chain of equations:

$VCD_{Obs} = R_{Obs} [...] = R_{RTM} [...]$

→ Thank you for the suggestion, and we agree that switching the second and last terms of Eq. (3) would be more coherent to the paragraph explaining the logical chain. We have modified Eq. (3) in the revised manuscript accordingly.

Eqn. (4): What is VCD_rtm? Should it be VCD_obs?

→ The fundamental equation for the MWP method is Eq. (1), and the subscripts (i.e., "rtm" and "obs") are introduced to consider the discrepancy between the real world and the model-world (i.e., RTM-world with assumed hypothetical atmospheric conditions). Our focus was particularly on dealing with the broad-band terms of reflectivity, which largely

depends on both the surface and aerosols characteristics and can affect the observed radiance ratio ($R$) significantly. Therefore, Eq. (1) can be written in two different version: one in model-world perspective (Eq. R1) and another in HIS-observed world perspective (Eq. R2).

$$VCD_{rtm} \; = \; R_{rtm} \cdot a + b \tag{R1}$$

$$VCD_{obs} \; = \; R_{obs} \cdot a + b \tag{R2}$$

The relationship between $VCD$ and $R$ (i.e., parameters "$a$" and "$b$" in Eqs. R1 and R2) is retrieved from model-world (Eq. R1). On the other hand, $VCD_{obs}$ in Eq. R2 is a "biased" estimate based on "HIS-observed" $R_{obs}$, which is affected by the unknown broad-band spectral dependency of reflectivity that has not been accounted for in the model-world, and the model-world-derived values of "$a$" and "$b$". Therefore, $VCD_{rtm}$ is the value we aim to retrieve, representing the "True" $VCD$, whereas the $VCD_{obs}$ is a value that we would assume to be "True" if we had not developed the MWP method. The MWP method essentially derives an "unbiased" $VCD$ based on the biased $R_{obs}$ values obtained from airborne HIS observations.

Line 231: In the MWP method: What makes A and B so special – wouldn't the whole equation also work for any wavelength pair independent of type (ascending or descending). Considering Table 2 and Figure 2 – would the wavelength pairs A2 and A3 not work similarly good as A2 and B2?

→ The advantage of using two different types of wavelength pairs can be explained through the final equation of the MWP method (i.e., Eq. 10). When the same type of wavelength pair (i.e., Type_A or Type_B) is used in Eq. (10), both the numerator and the denominator of the first term on the right-hand side are likely to be excessively small in absolute values. Considering the relatively low sensitivity of the $R$ values corresponding to the NO$_2$ VCD changes, which in turn results in large variability of "$a$" and "$b$" values due to the measurement uncertainties, denominator and numerator with the small absolute value can significantly amplify the effect of uncertainties. Therefore, it is necessary to ensure that $a_A$ and $a_B$ have different signs (as well as $b_A$ and $b_B$) to ensure sufficiently large values for the denominator and numerator in Eq. (10). We acknowledge that we have not adequately elaborated on the details about the virtue of using two

different types of wavelength pairs. In the revised manuscript, we have added brief explanations to address this point (lines 157–160, and 179).

Line 267: The authors describe the use of NCEP data set as the initial conditions for the WRF model. Since ERA5 data set was used as input for the RTM simulations, wouldn't it be more consistent to use the same data for the WRF model.

→ The use of CMAQ model outputs was limited to constructing a pool of $NO_2$ vertical profiles. This allowed us to reasonably estimate the model (i.e., RTM)-input $NO_2$ vertical profile based only on parameters such as PBLH and $NO_2$ VCD. Therefore, we agree that using the same ERA5 data as an initial boundary condition of the CTM would undoubtedly be more or less helpful, but using the NCEP/FNL data would not significantly affect the results. Therefore, we have run the model favorably considering the feasibility.

Line 273: How strong would variations in the assumed stratospheric $NO_2$ affect the retrieved $NO_2$ VCD?

→ The variation of stratospheric $NO_2$, or its deviation from the assumed US standard atmosphere $NO_2$ stratospheric profile, will have minimal effect on the $NO_2$ VCD retrieved from the airborne HIS observations. Applying the spectral scale factor (SF) calibration has an equivalent effect to using the reference spectrum obtained from each respective airborne HIS observation. Since stratospheric $NO_2$ has limited short-term variability (i.e., < a couple of hours), most of the stratospheric signal will be canceled out.

Table 4: The uncertainties introduced into the RTM calculations by the assumptions given in 3.2 (ERA5 pressure, temperature, mixing ratios and CTM data for $NO_2$ vertical profiles) seem to be missing in the error estimation description. A sensitivity study of the quantities used in the RTM calculations would be helpful.

→ We sincerely appreciate your comment and have contemplated the possible additional source of error in our retrieval, focusing on the aspects you suggested. It is certainly possible that the pressure and temperature profiles can affect the air density profile and, consequently, the air mass factor and the retrieved $NO_2$ VCDs. However, it has been known

that the temperature and pressure profile variabilities are not significant enough to affect the AMF considerably (Boersma et al., 2004), whereas those profiles can affect the effective temperature of $NO_2$ and can take part in the AMF temperature correction factor (Lorente et al., 2017). The temperature correction to the AMF can be notable, especially over industrial regions with large tropospheric $NO_2$ variability, attributed to significant temperature dependency of $NO_2$ absorption in VIS bands (Spinei et al., 2014). However, the primary contributor determining the effective temperature is rather the $NO_2$ vertical profile (shape); hence, neither the temperature nor pressure profile should draw attention in terms of uncertainty estimates.

Without parameterization, it is complicated and even more inherently limited to make sensitivity tests upon vertical profile shape (i.e., $NO_2$, pressure, temperature). For instance, Boersma et al. (2018) assumed uncertainty contribution from satellite $NO_2$ vertical profile shape is approximately 10% in Quality Assurance for the Essential Climate Variables (QA4ECV) project, while Tack et al. (2019) assumed it as 7 ~ 10 % for the airborne observation using hyperspectral imaging sensors. Therefore, we parameterized $NO_2$ vertical profiles with the PBLH, using the pool of $NO_2$ vertical profiles and their corresponding PBLH conditions simulated with CTM (i.e., CMAQ). We further assumed that the aerosol profile also conforms with the $NO_2$ profile, and the sensitivity test has been conducted to the PBLH as an alternative.

We acknowledge that there can be a caveat on the sensitivity test and consequential error estimations. For example, such assumptions as aerosol properties (i.e., vertical profile, optical properties), $NO_2$ vertical profile, effective temperature of $NO_2$, and presence of small-localized clouds can also comprise part of the total uncertainty. Therefore, we decided to mention at least these potential additional sources of uncertainties in the revised manuscript, and have modified it accordingly (please refer to lines 438–445 and 661–663 in the revised manuscript).

References

Boersma, K. F., Eskes, H. J., and Brinksma, E. J.: Error analysis for tropospheric $NO_2$ retrieval from space, Journal of Geophysical Research, 109 (D04311), :10.1029/2003JD003962, 2004.

Boersma, K. F., Eskes, H. J., Richter, A., De Smedt, I., Lorente, A., Beirle, S., van Geffen, J. H. G. M., Zara, M., Peters, E., Van Roozendael, M., Wagner, T., Maasakkers, J. D., van der A, R. J., Nightingale, J., De Rudder, A., Irie, H., Pinardi, G., Lambert, J.-C., and Compernolle, S. C.: Improving algorithms and uncertainty estimates for satellite $NO_2$ retrievals: results from the quality assurance for the essential climate variables (QA4ECV) project, Atmos. Meas. Tech., 11, 6651–6678, https://doi.org/10.5194/amt-11-6651-2018, 2018.

Lorente, A., Folkert Boersma, K., Yu, H., Dörner, S., Hilboll, A., Richter, A., Liu, M., Lamsal, L. N., Barkley, M., De Smedt, I., Van Roozendael, M., Wang, Y., Wagner, T., Beirle, S., Lin, J.-T., Krotkov, N., Stammes, P., Wang, P., Eskes, H. J., and Krol, M.: Structural uncertainty in air mass factor calculation for $NO_2$ and HCHO satellite retrievals, Atmos. Meas. Tech., 10, 759–782, https://doi.org/10.5194/amt-10-759-2017, 2017.

Spinei, E., Cede, A., Swartz, W. H., Herman, J., and Mount, G. H.: The use of $NO_2$ absorption cross section temperature sensitivity to derive $NO_2$ profile temperature and stratospheric–tropospheric column partitioning from visible direct-sun DOAS measurements, Atmos. Meas. Tech., 7, 4299–4316, https://doi.org/10.5194/amt-7-4299-2014, 2014.

Tack, F., Merlaud, A., Meier, A. C., Vlemmix, T., Ruhtz, T., Iordache, M.-D., Ge, X., van der Wal, L., Schuettemeyer, D., Ardelean, M., Calcan, A., Constantin, D., Schönhardt, A., Meuleman, K., Richter, A., and Van Roozendael, M.: Intercomparison of four airborne imaging DOAS systems for tropospheric $NO_2$ mapping – the AROMAPEX campaign, Atmos. Meas. Tech., 12, 211–236, https://doi.org/10.5194/amt-12-211-2019, 2019.

**Technical revisions**

Line 90: "80 % of the staturation level, and the CCD" -> "80 % of the saturation level, and thus the CCD"

→ Thank you for the advice. We have added "thus" in the corresponding sentence.

Line 186: "the wavelength pair" -> "each wavelength pair"

→ Thank you for the advice. We have replaced "the" with "each" in the corresponding sentence.

Line 209: "Then the Eq. (2) can be reformulated by replacing [...]" – I think it should be Eq. 1.

→ Thank you for pointing out our mistake. It should be "first equation of the simultaneous equations of Eq. (3)", instead of "Eq. (2)". We have modified them in the revised manuscript (line 216).

**Suggested/required improvements for the Figures**

All figures

Tooltips are in Korean

→ This issue arises whenever we try to convert our manuscript from MS Word format (i.e., .docx file) to PDF. We have spent hours trying to fix this problem, following the instructions we found online, and we no longer see the issue in the preprint manuscript when opened with our PDF viewer. Nevertheless, we apologize for any inconvenience caused. The tooltips on the figures were added inadvertently and do not contain essential information. We believe that this will be resolved in the final version of the published article since we will submit the figures as separate ".png" files.

Figure 1

The figure description should mention what red/gray/blue boxes stand for – I guess its red: input data, gray: retrieval step, blue: result?!

→ The (blue) oval is for the start/end of a flowchart, the (red) parallelogram represents input or output, and the (gray) rectangle represents a process. The shape of the boxes (i.e., oval, parallelogram, and rectangle) are the convention used for the flowchart, and colors are differentiated to enhance visibility. We have not added any explanations for the shape of the boxes because we followed the convention of the flowcharts.

I don't think that there should be an arrow from "NO₂ VCD retrieval" to "HIS raw spectra". I guess the NO₂ VCD retrieval is the whole thing depicted in this figure? Suggestion: Remove the arrow.

→ As explained in response to the previous question, we tried to adhere to the convention of the flowcharts. We kindly ask for your understanding regarding the decision we made to keep the figure as it was in the original manuscript.

Figure and description make clear that there is a strong dependency on input variables. How independent is the method and what is the uncertainty caused by errors in the assumption? (see also comment/minor revision on Table 4)

→ We are not certain whether we have comprehended your comment correctly, but we have discussed about the uncertainties caused by uncertainties of the input variables in Sect. 3.4. However, we acknowledge that we have not discussed about the assumptions (i.e., NO₂ vertical profile shape), and have added some discussions accordingly in the revised manuscript (lines 438–445).

Figure 3

Summarizing the figure in the text when referred to would be quite beneficial in understanding the content a lot faster – also some reasoning why some dependencies are found would help:

→ We have tried to elaborate the results shown in Fig. 3 (i.e., propagated uncertainties from the uncertainty range of each input parameter shown in Table 4), and have explained them to a certain extent in descending order based on their relative contribution to the $Q_{rel}$ uncertainty (lines 400–435 in the revised manuscript). We are not sure what your suggestion is in this comment precisely, but we have tried to answer and address the detailed comments below.

None of the investigated uncertainties show a dependency on PBLH, altitude, solar zenith angle, viewing zenith angle or relative azimuth angle.

→ There surely are uncertainties propagated from the PBLH, altitude, solar zenith angle, viewing zenith angle, and relative azimuth angle. However, their magnitude is much smaller than that has been posed by the uncertainties in SF calibration, wavelength shift,

or instrument noise. We believed that the uncertainty sources with relatively significant contributions to the overall uncertainty of retrieved $NO_2$ VCD should be highlighted in Fig. 3, and as the drawback, the parameters with relatively minor contributions to total uncertainty were not discernible in Fig. 3 (although they are presented in the figure). As discussed in the original manuscript, the reason for the limited uncertainty introduced by those parameters is mainly attributed to the high-level certainty of those parameters (i.e., altitude, solar zenith angle, viewing zenith angle, relative azimuth angle) associated with environmental conditions of research flights carefully selected to minimize the sensitivity of those parameters (i.e., solar zenith angle, viewing zenith angle).

The uncertainty contribution of the Albedo increases for small albedos (< 0.15) – is this caused by an absolute uncertainty of the albedo which becomes relatively large for small values of the albedo?

→ Yes, that is the likely cause of the tendency shown in Fig. 3c. Please refer to the lines 424–425 in the revised manuscript.

The uncertainty contribution of the Albedo and the SF uncertainty slightly increase with increasing $NO_2$ VCDs. Is this connected to the probability of the SF is contaminated by $NO_2$ increases with $NO_2$ VCDs? Why does the albedo uncertainty depend on the $NO_2$ VCD?

→ We believe that the tendency of increasing uncertainties in $Q_{rel}$ ($\sigma_{Q_{rel}}$) incurred from the albedo uncertainty as $NO_2$ VCD increase (shown in Fig. 3a) is related to the $NO_2$ vertical profile with a greater portion near the surface (i.e., within the PBLH) under higher $NO_2$ VCD conditions. Therefore, the near-surface sensitivity (i.e., Jacobian) shift due to albedo change can result in greater AMF change, and consequentially the $NO_2$ VCD, under higher $NO_2$ VCD conditions. This has been explained in the original manuscript and can also be found in the revised manuscript (lines 425–430).

The $\sigma_Q$ induced from SF calibration uncertainties should have no dependency on $NO_2$ VCD. However, Fig. 3 shows $\sigma_{Q_{rel}}$, which can be calculated by dividing $\sigma_Q$ by the corresponding $Q$ value. The $Q$ value decreases as the $NO_2$ VCD increases by its definition (i.e., $Q = {R_A}/{R_B}$), since $R_A$ decreases and $R_B$ increases as the $NO_2$ VCD increases. Consequently, $\sigma_{Q_{rel}}$ caused by SF calibration uncertainty becomes greater under higher $NO_2$ VCD conditions. Moreover, the dependency of SF calibration uncertainty

propagated to $Q_{rel}$ according to NO₂ VCD is much smaller than that of albedo uncertainty, while the reason for this dependency is more like a technical issue caused by showing uncertainty propagated to $Q_{rel}$ instead of $Q$ and not the physical problem to be explained. Therefore, we have decided not to make detailed explanations regarding the SF calibration uncertainty propagated to $Q_{rel}$ having a dependency on NO₂ VCD in our manuscript.

The figure could generally use a more consistent description mentioning all used abbreviations. The authors should provide more information on "wvl" and "Instrument noise".

→ We speculate that you might be referencing the previous version of the manuscript (initial submission). We have revised them in our preprint version manuscript, and changed "wvl" to "Wavelength shift" and added explanatory "SNR" within the following parenthesis of "Instrument noise" in Fig. 3 legend.

Figure 4

The figure is generally quite hard to read/understand. Here are some suggestions:

→ We appreciate your advice, and we discussed your suggestion thoroughly and made some modifications in the revised manuscript. Please refer to our specific response to each comment below.

When using text within the figures only use "white" as a text color.

→ When the color of text within the figures is fixed to "white", some of them are hard to distinguish from the background TROPOMI image. We have tested with several different combinations of text colors and text outline colors and have modified the figure in the revised manuscript with the "best" combination we believe. We hope it is satisfactory to you.

There is no description of red boxes or blue circles – maybe unify marked regions with one color (e.g. white)

→ We tried to use vivid colors for better discernibility, but we acknowledge that the red boxes/blue circles without description might be confusing. We have changed the red boxes in the upper left figure to white in our revised manuscript, while keeping the blue

circles since it is consistent with later figures (i.e., Figs. 5~7). We found that the description about the blue circle is missing in the Fig. 4 caption, so have added them in the revised manuscript.

The legend in the upper left panel is valid for all panels – it should be moved out of that panel, so this becomes clear.

→ Figure 4 is not intended to separate and label all four panels shown as a subplot (i.e., Fig. 4a, 4b, …. etc.), but to show all four panels as a single figure showing the general $NO_2$ VCD distribution of the target domains. Moreover, as you commented below, the upper left panel stands for the "label legend" of other three panels. Therefore, we believe that keeping the marker legend within the upper left figure makes sense. Keeping this legend within the upper left panel is also preferable since it can ensure the figure to be as large as possible in the limited space. Nevertheless, we acknowledge that readers might feel confused, so have added an explicit statement to refer to the legend in the upper left figure for the symbols and acronyms of the industrial point sources in the Fig. 4 caption of the revised manuscript.

The individual panels should be named. Right now, the upper left figure is the "label legend" for all 4 panels – maybe it is enough to show a bigger version of the upper left figure?

→ Since the target domain that has been investigated in this study is a relatively small area in South Korea, it would be hard to mark all the concentrated point sources within these areas in a discernible manner. Moreover, we wanted to avoid potential misleading that all the industrial point sources in Korea are located within the target domains (i.e., Chungnam, Jecheon, and Pohang) by showing a single enlarged version of the upper left figure and marking only the industrial point sources within these target domains. Therefore, we decided to keep the general outline of the figure.

Panels (a), (b) and (c) look peculiar. What gridding routine was used for the satellite data and which filters (e.g. clouds) were applied? It is not fully clear which data were averaged to obtain the displayed $NO_2$ distribution.

→ Figures 4, 5a, 6a, and 7a are the visualization of downscaled TROPOMI $NO_2$ VCD$_{Total}$ (entitled "VCD$_{summed}$" in TROPOMI product) composites from TROPOMI swaths in October and November from 2019 to 2022. First, the downscaled grid has been pre-determined

for each figure domain (i.e., 0.25°×0.25° for Korea, 0.1°×0.1° for the rest of the domains). For every downscaled grid point and TROPOMI swath, TROPOMI pixels located within approximately ± 2 pixel width from the downscaled grid point were selected, and the distances between the downscaled grid point and the selected TROPOMI pixels' center were calculated. Using the inverse of those distances as a weight, the weighted average was taken as a $NO_2$ VCD value for a single downscaled grid point in a typical TROPOMI swath. Finally, the average of the $NO_2$ VCD values corresponding to each downscaled grid point were taken and has been visualized as shown in the figures. It is noteworthy that all TROPOMI $NO_2$ VCD swath data were screened with quality flags (qa_value > 0.75).

The details of the downscaling methodology (as shown above) were omitted in the manuscript because those figures (i.e., Figs. 4, 5a, 6a, and 7a) were intended to show general $NO_2$ VCD distribution in each respective target domain at a similar time of year, without elaborating with detailed discussions. The figures you have pointed out may appear peculiar, attributed to the limited availability of TROPOMI swath data over the target domains in October and November. We now perceive that some readers might find figures puzzling. Consequently, we added more details to the revised manuscript (Fig. 4 caption). We appreciate your comment.

Figures 5, 6, 7

The results of the airborne measurements are hard to read.

Generally, the same problems as in Figure 4 apply.

→ We have modified the figures to be consistent with the modifications made in Fig. 4.

The figures need to be larger/better resolved to be readable.

→ We apologize for the inconvenience caused by the limited resolution of the figures. The original versions of the figures, as well as the figures in the manuscript in MS Word format, display sufficiently high resolution (we have adhered to the AMT publication guidelines, and all figures have a DPI greater than 300). However, the resolution of the figures somehow decreases when we use the MS PDF converter to convert the MS Word file to PDF. We believe this issue will be resolved in the final version of the published article, as the figures will be replaced with separately submitted high-resolution ".png" files.

Are the shown days cloud-free or are cloud data just not flagged? Interpretation of $NO_2$ columns retrieved from satellite instruments are difficult to interpret in cloud contaminated scenes, it should be made clear whether the measurement days were cloud free or a cloud flag needs to be applied.

→ Please refer to our response above to the comment about cloud-screening of TROPOMI data (the last question in the general/major revision section). As a brief reminder, we have conducted the research flight in a cloud-free environment (i.e., location, time), and should also be the collocated TROPOMI swaths cloud-free. We have double-checked whether the collocated TROPOMI pixels are cloud-contaminated, and found that none of the TROPOMI pixels had effective cloud fraction exceeding the QA criteria (CF < 0.3). Moreover, all the TROPOMI data used in this study complied with the QA criteria suggested in the Product User Manual (Eskes et al., 2022).

References

Eskes, H. J., van Geffen, J., Boersma, F., Eichmann, K.-U., Apituley, A., Pedergnana, M., Sneep, M., Veefkind, J. P., and Loyola, D.: Sentinel-5 precursor/TROPOMI Level 2 Product User Manual Nitrogen Dioxide, V.4.1.0 (S5P-KNMI-L2-0021-MA), 2022.

Figure 8

If the TROPOMI data are not cloud filtered, this should also be done here.

→ We have used cloud-filtered TROPOMI data throughout the manuscript. Please refer to the AR to the RC1 above.

**Again, we appreciate for your valuable comments and suggestions.**

===End of Document===